# Reinforcement Learning for Node Selection in Branch-and-Bound

## Abstract

A big challenge in branch and bound lies in identifying the optimal node within the search tree from which to proceed. Current state-of-the-art selectors utilize either hand-crafted ensembles that automatically switch between naive sub-node selectors, or learned node selectors that rely on individual node data. We propose a novel bi-simulation technique that uses reinforcement learning (RL) while considering the entire tree state, rather than just isolated nodes. To achieve this, we train a graph neural network that produces a probability distribution based on the path from the model's root to its "to-be-selected" leaves. Modelling node-selection as a probability distribution allows us to train the model using state-of-the-art RL techniques that capture both intrinsic node-quality and node-evaluation costs. Our method induces a high quality node selection policy on a set of varied and complex problem sets, despite only being trained on specially designed, synthetic travelling salesmen problem (TSP) instances. Using such a fixed pretrained policy shows significant improvements on several benchmarks in optimality gap reductions and per-node efficiency under strict time constraints.

## 1 Introduction

The optimization paradigm of mixed integer programming plays a crucial role in addressing a wide range of complex problems, including scheduling (Bayliss et al., 2017), process planning (Floudas & Lin, 2005), and network design (Menon et al., 2013). A prominent algorithmic approach employed to solve these problems is *branch-and-bound (BnB)*, which recursively subdivides the original problem into smaller sub-problems through variable branching and pruning based on inferred problem bounds. BnB is also one of the main algorithms implemented in SCIP (Bestuzheva et al., 2021a;b), a state-of-the art mixed integer linear and mixed integer nonlinear solver.

An often understudied aspect is the node selection problem, which involves determining which nodes within the search tree are most promising for further exploration. This is due to the intrinsic complexity of understanding the emergent effects of node selection on overall performance for human experts. Contemporary methods addressing the node selection problem typically adopt a per-node perspective (Yilmaz & Yorke-Smith, 2021; He et al., 2014; Morrison et al., 2016), incorporating varying levels of complexity and relying on imitation learning (IL) from existing heuristics (Yilmaz & Yorke-Smith, 2021; He et al., 2014). However, they fail to fully capture the rich structural information present within the branch-and-bound tree itself.

We propose a novel selection heuristic that leverages the power of bi-simulating the branch-and-bound tree with a neural network-based model and that employs reinforcement learning (RL) for heuristic training, see Fig. 1. To do so, we reproduce the SCIP state transitions inside our neural network structure (bi-simulation), which allows us to take advantage of the inherent structures induced by branch-and-bound. By simulating the tree and capturing its underlying dynamics we can extract valuable insights that inform the RL policy, which learns from the tree's dynamics, optimizing node selection choices over time.

We reason that RL specifically is a good fit for this type of training as external parameters outside the pure quality of a node have to be taken into account. For example, a node $A$ might promise a significantly bigger decrease in the expected optimality gap than a second node $B$, but node $A$ might take twice as long to evaluate, making $B$ the "correct" choice despite its lower theoretical utility. By

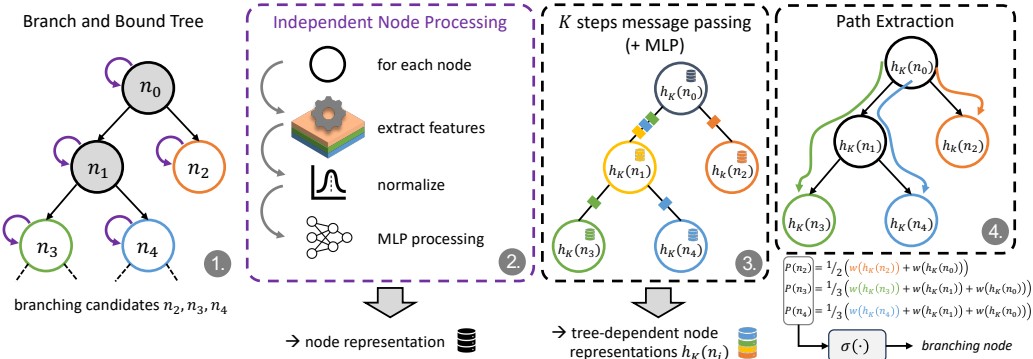

Figure 1: Our method: (1) SCIP solves individual nodes and executes existing heuristics. (2) Features are extracted from every branch-and-bound node and sent to individual normalization and embedding. (3) The node embeddings are subject to $K$ steps of GNN message passing on the induced tree-structure. (4) Based on the node embeddings, we generate root-to-leaf paths, from which we sample the next node. The resulting node is submitted to SCIP and we return to step 1.

incorporating the bi-simulation technique, we can effectively capture the intricate interdependencies of nodes and propagate relevant information throughout the tree.

## 2 BRANCH AND BOUND

BnB is one of the most effective methods for solving mixed integer programming (MIP) problems. It recursively solves relaxed versions of the original problem, gradually strengthening the constraints until it finds an optimal solution. The first step relaxes the original MIP instance into a tractable subproblem by dropping all integrality constraints such that the subproblem can later be strictified into a MIP solution. For simplicity, we focus our explanation to the case of mixed integer linear programs (MILP) while our method theoretically works for any type of constraint allowed in SCIP (see nonlinear results in Sec. 5.3.2, and (Bestuzheva et al., 2023). Concretely a MILP has the form

$$P_{\text{MILP}} = \min\{c_1^T x + c_2^T y | Ax + By \geq b, y \in \mathbb{Z}^n\}, \tag{1}$$

where $c_1$ and $c_2$ are coefficient vectors, $A$ and $B$ are constraint matrices, and $x$ and $y$ are variable vectors. The integrality constraint $y \in \mathbb{Z}^n$ requires $y$ to be an integer. In the relaxation step, this constraint is dropped, leading to the following simplified problem:

$$P_{\text{relaxed}} = \min\{c_1^T x + c_2^T y | Ax + By \geq b\}. \tag{2}$$

Now, the problem becomes a linear program without integrality constraints, which can be exactly solved using the Simplex (Dantzig, 1982) or other efficient linear programming algorithms.

After solving the relaxed problem, BnB proceeds to the branching step: First, a non-integral $y_i$ is chosen. The branching step then derives two problems: The first problem (Eq. 3) adds a lower bound to variable $y_i$, while the second problem (Eq. 4) adds an upper bound to variable $y_i$. These two directions represent the rounding choices to enforce integrality for $y_i$:[1]

$$P_{\text{relaxed}} \cup \{y \leq c\} = \min\{c_1^T x + c_2^T y | Ax + By \geq b, y_i \leq \lfloor c \rfloor\} \tag{3}$$

$$P_{\text{relaxed}} \cup \{y \geq c+1\} = \min\{c_1^T x + c_2^T y | Ax + By \geq b, y_i \geq \lceil c \rceil\} \tag{4}$$

The resulting decision tree, with two nodes representing the derived problems can now be processed recursively. However, a naive recursive approach exhaustively enumerates all integral vertices, leading to an impractical computational effort. Hence, in the bounding step, nodes that are deemed worse than the currently known best solution are discarded. To do this, BnB stores previously found solutions which can be used as a lower bound to possible solutions. If a node has an upper bound larger than a currently found integral solution, no node in that subtree has to be processed.

---

[1]There are non-binary, "wide" branching strategies which we will not consider here explicitly. However, our approach is flexible enough to allow for arbitrary branching width. See Morrison et al. (2016) for an overview.

The interplay of these three steps—relaxation, branching, and bounding—forms the core of branch-and-bound. It enables the systematic exploration of the solution space while efficiently pruning unpromising regions. Through this iterative process, the algorithm converges towards the optimal solution for the original MIP problem, while producing exact optimality bounds at every iteration.

## 3 RELATED WORK

While variable selection through learned heuristics has been studied a lot (see e.g. Parsonson et al. (2022) or Etheve et al. (2020)), learning node selection, where learned heuristics pick the best node to continue, have only rarely been addressed in research. We study learning algorithms for node selection in state-of-the-art branch-and-cut solvers. Prior work that learns such node selection strategies made significant contributions to improve the efficiency and effectiveness of the optimization.

Notably, many approaches rely on per-node features and Imitation Learning (IL). Otten & Dechter (2012) examined the estimation of subproblem complexity as a means to enhance parallelization efficiency. By estimating the complexity of subproblems, the algorithm can allocate computational resources more effectively. Yilmaz & Yorke-Smith (2021) employed IL to directly select the most promising node for exploration. Their approach utilized a set of per-node features to train a model that can accurately determine which node to choose. He et al. (2014) employed support vector machines and IL to create a hybrid heuristic based on existing heuristics. By leveraging per-node features, their approach aimed to improve node selection decisions. While these prior approaches have yielded valuable insights, they are inherently limited by their focus on per-node features.

Labassi et al. (2022) proposed the use of Siamese graph neural networks, representing each node as a graph that connects variables with the relevant constraints. Their objective was direct imitation of an optimal diving oracle. This approach facilitated learning from node comparisons and enabled the model to make informed decisions during node selection. However, the relative quality of two nodes cannot be fully utilized to make informed branching decisions as interactions between nodes remain minimal (as they are only communicated through a final score and a singular global embedding containing the primal and dual estimate). While the relative quality of nodes against each other is used, the potential performance is limited as the overall (non-leaf) tree structure is not considered.

Another limitation of existing methods is their heavy reliance on pure IL that will be constrained by the performance of the existing heuristics. Hence, integrating RL to strategically select nodes holds great promise. This not only aims for short-term optimality but also for the acquisition of valuable information to make better decisions later in the search process. This is important as node selection not only has to model the expected decrease in the optimality gap, but also has to account for the time commitment a certain node brings with it as not all nodes take equal amount of time to process.

## 4 METHODOLOGY

We combine two major objectives: a representation that effectively captures the inherent structures and complexities of the branch-and-bound tree, see Sec. 4.1, and an RL policy trained to select nodes (instead of an heuristic guided via RL), see Sec. 4.2. To do this, we view node-selection as a probabilistic process where different nodes $n_k$ can be sampled from a distribution $\pi$: $n_k \sim \pi(n_k|s_o)$, where $s_o$ is the state of the branch-and-bound optimizer. Optimal node selection can now be framed as learning a node-selection distribution $\pi$ that, given some representation of the optimizer state $s_o$, selects the optimal node $n_i$ to maximize a performance measure. A reasonable performance measure, for instance, captures the achieved performance against a well-known baseline. In our case, we choose our performance measure, i.e., our reward, to be

$$r = -\left(\frac{gap(node\ selector)}{gap(scip)} - 1\right) \tag{5}$$

The aim is to decrease the optimality gap achieved by our node-selector at the end of the solve (i.e., *gap(node selector)*), normalized by the results achieved by the existing state-of-the-art node-selection methods in the SCIP (Bestuzheva et al., 2021a) solver at the end of the solve (i.e., *gap(scip)*). We further shift this performance measure, such that any value $> 0$ corresponds to our selector being superior to existing solvers, while a value $< 0$ corresponds to our selector being

worse, and clip the reward objective between $(1, -1)$ to ensure equal size of the reward and "punishment" ranges as is commonly done in prior work (see, e.g. Mnih et al. (2015). In general, finding good performance measurements that areis difficult (see Sec. 5). For training we can circumvent a lot of the downsides of this reward formulation, like divide-by-zero or divide-by-infinity problems, by simply sampling difficult, but tractable problems (see Sec. 4.3).

## 4.1 TREE REPRESENTATION

To address the first objective, we propose a novel approach that involves bi-simulating the existing branch-and-bound tree using a graph neural network (GNN). This entails transforming the tree into a graph structure, taking into account the features associated with each node (for a full list of features we used, see Appendix C). We ensure that the features stay at a constant size, independent from, e.g., the number of variables and constraints, to enable efficient batch-evaluation of the entire tree.

In the reconstructed GNN, the inputs consist of the features of the current node, as well as the features of its two child nodes. Pruned or missing nodes are replaced with a constant to maintain the integrity of the graph structure. This transformation enables us to consider subtrees within a predetermined depth limit, denoted as $K$, by running $K$ steps of message passing. This approach allows us to balance memory and computational requirements across different nodes, while preventing the overload of latent dimensions in deep trees.

For our graph processing GNN, we use a well understood method known as "Message Passing" across nodes. For all nodes, this method passes information from the graph's neighborhood into the node itself, by first aggregating all information with a permutation invariant transform (e. g., computing the mean across neighbors), and then updating the node-state with the neighborhood state. In our case, the (directed) neighborhood is simply the set of direct children (see Fig. 1). As message passing iterates, we accumulate increasing amounts of the neighborhood, as iteration $t + 1$ utilizes a node-embedding that already has the last $t$ steps aggregated. Inductively, the message passing range directly correlates with the number of iterations used to compute the embeddings.

Concretely, the internal representation can be thought of initializing $h_0(n) = x(n)$ (with $x(n)$ being the feature associated with node $n$) and then running $K$ iterations jointly for all nodes $n$:

$$h_{t+1}(n) = h_t(n) + \text{emb}\left(\frac{h_t(\text{left}(n)) + h_t(\text{right}(n))}{2}\right), \tag{6}$$

where $\text{left}(n)$ and $\text{right}(n)$ are the left and right children of $n$, respectively, $h_t(n)$ is the hidden representation of node $n$ after $t$ steps of message passing, and $\text{emb}$ is a function that takes the mean hidden state of all embeddings and creates an updated node embedding.

## 4.2 RL FOR NODE SELECTION

While the GNN model is appealing, it is impossible to train using contemporary imitation learning techniques, as the expert's action domain (i.e., leaves) may not be the same as the policy's action domain, meaning that the divergence between these policies is undefined. Instead, we phrase our node selection MDP as a the (state, action, reward) triple of (BnB tree, selectable leaves, reward $r$) (where $r$ is defined according to Eq. 5) and use RL techniques to solve this problem. Using the final node-representation $h_K(n)$ we can derive a value for every node $V(h_K(n))$ and a weight $W(h_K(n))$ to be used by our RL agent. Specifically we can produce a probability distribution of node-selections (i. e., our actions) by computing the expected weight across the unique path from the graph's root to the "to-be-selected" leaves. We specifically consider the expectation as to not bias against deeper or shallower nodes. This definition allows us to have a global view on the node-selection probability, despite the fact that we only perform a fixed number of message-passing iterations to derive our node embeddings. Concretely, let $n$ be a leaf node in the set of candidate nodes $\mathscr{C}$, also let $P(r, n)$ be the unique path from the root $r$ to the candidate leaf node, with $|P(r, n)|$ describing its length. We define the expected path weight $W'(n)$ to a leaf node $n \in \mathscr{C}$ as

$$W'(n) = \frac{1}{|P(r, n)|} \sum_{u \in P(r, n)} W(h_K(u)). \tag{7}$$

Selection now is performed in accordance to sampling from a selection policy $\pi$ induced by

$$\pi(n | \text{tree}) = \text{softmax}\left(\{W'(n) | \forall n \in \mathscr{C}\}\right). \tag{8}$$

Intuitively, this means that we select a node exactly if the expected utility along its path is high. Note that this definition is naturally self-correcting as erroneous over-selection of one subtree will lead to that tree being completed, which removes the leaves from the selection pool $\mathscr{C}$.

By combining the bi-simulation technique, the GNN representation, and the computation of node probabilities, we establish a framework that enables distributional RL for node selection. We consider proximal policy optimization (PPO) (Schulman et al., 2017) for optimizing the node-selection policy. For its updates, PPO considers the advantage $A$ of the taken action in episode $i$ against the current action distribution. Intuitively, this amounts to reducing the frequency of disadvantageous actions, while increasing the frequency of high quality actions. We choose the generalized advantage estimator (GAE) (Schulman et al., 2015), which interpolates between an unbiased but high variance Monte Carlo estimator, and a biased, low variance estimator. For the latter we use a value function $V(s)$, which we implemented similarly to the policy-utility construction above:

$$Q(n|s) = \frac{\tilde{Q}(n|s)}{|P(r,n)|} \tag{9}$$

$$\tilde{Q}(n|s) = \tilde{Q}(\text{left child}|s) + \tilde{Q}(\text{right child}|s) + q(h_n|s) \tag{10}$$

$$V(s) = \{\max Q(n) \mid \forall n \in \mathscr{C}\} \tag{11}$$

where $q(h_n)$ is the per-node estimator, $\tilde{Q}$ the unnormalized Q-value, and $\mathscr{C}$ is the set of open nodes as proposed by the branch-and-bound method. Note that this representation uses the fact that the value function can be written as the maximal Q-value: $V(s) = \max_{a \in A} Q(a|s)$.

This method provides low, but measurable overhead compared to existing node selectors, even if we discount the fact that our Python-based implementation is vastly slower than SCIP's highly optimized C-based implementations. Hence, we focus our model on being efficient at the beginning of the selection process, where good node selections are exponentially more important as finding more optimal solutions earlier allows to prune more nodes from the exponentially expanding search tree. Specifically we evaluate our heuristic at every node for the first 250 selections, then at every tenth node for the next 750 nodes, and finally switch to classical selectors for the remaining nodes.[2]

### 4.3 DATA GENERATION & AGENT TRAINING

In training MIPs, a critical challenge lies in generating sufficiently complex training problems. First, to learn from interesting structures, we need to decide on some specific problem, whose e. g., satisfiability is knowable as generating random constraint matrices will likely generate empty polyhedrons, or polyhedrons with many eliminable constraints (e.g., in the constraint set consisting of $c^T x \leq b$ and $c^T x \leq b + \rho$ with $\rho \neq 0$ one constraint is always eliminable). This may seem unlikely, but notice how we can construct aligned $c$ vectors by linearly combining different rows (just like in LP-dual formulations). In practice, selecting a sufficiently large class of problems may be enough as during the branch-and-cut process many sub-polyhedra are anyways being generated. Since our algorithm naturally decomposes the problem into sub-trees, we can assume any policy that performs well on the entire tree also performs well on sub-polyhedra generated during the branch-and-cut.

For this reason we consider the large class of Traveling Salesman Problem (TSP), which itself has rich use-cases in planning and logistics, but also in optimal control, the manufacturing of microchips and DNA sequencing (see Cook et al. (2011)). The TSP problem consists of finding a round-trip path in a weighted graph, such that every vertex is visited exactly once, and the total path-length is minimal (for more details and a mathematical formulation, see Appendix A)

For training, we would like to use random instances of TSP but generating them can be challenging. Random sampling of distance matrices often results in easy problem instances, which do not challenge the solver. Consequently, significant effort has been devoted to devising methods for generating random but hard instances, particularly for problems like the TSP, where specific generators for challenging problems have been designed (see Vercesi et al. (2023) and Rardin et al. (1993)).

However, for our specific use cases, these provably hard problems may not be very informative as they rarely contain efficiently selectable nodes. For instance, blindly selecting knapsack instances

---

[2]This accounts for the "phase-transition" in MIP solvers where optimality needs to be proved by closing the remaining branches although the theoretically optimal point is already found (Morrison et al., 2016). Note that with a tuned implementation we could run our method for more nodes, where we expect further improvements.

according to the Merkle-Hellman cryptosystem (Merkle & Hellman, 1978), would lead to challenging problems, but ones that are too hard to provide meaningful feedback to the RL agent.

To generate these intermediary-difficult problems, we adopt a multi-step approach: We begin by generating random instances and then apply some mutation techniques (Bossek et al., 2019) to introduce variations, and ensure diversity within the problem set. From this population of candidate problems, we select the median optimality-gap problem. The optimality gap, representing the best normalized difference between the lower and upper bound for a solution found during the solver's budget-restricted execution, serves as a crucial metric to assess difficulty. This method is used to produce 200 intermediary-difficulty training instances

To ensure the quality of candidate problems, we exclude problems with more than $100\%$ or zero optimality gap, as these scenarios present challenges in reward assignment during RL. To reduce overall variance of our training, we limit the ground-truth variance in optimality gap. Additionally, we impose a constraint on the minimum number of nodes in the problems, discarding every instance with less than 100 nodes. This is essential as we do not expect such small problems to give clean reward signals to the reinforcement learner.

## 5 EXPERIMENTS

For our experiments we consider the instances of TSPLIB (Reinelt, 1991) and MIPLIB (Gleixner et al., 2021) which are one of the most used datasets for benchmarking MIP frameworks and thusly form a strong baseline to test against. We further test against the UFLP instance generator by (Kochetov & Ivanenko, 2005), which specifically produces instances hard to solve for branch-and-bound, and against MINLPLIB (Bussieck et al., 2003), which contains *mixed integer nonlinear programs*, to show generalization towards very foreign problems. The source code for reproducing our experiments will be made publicly available (see supplementary material).

### 5.1 BASELINES

We run both our method and SCIP for 45s.[3] We then filter out all runs where SCIP has managed to explore less than 5 nodes, as in these runs we cannot expect even perfect node selection to make any difference in performance. If we included those in our average, we would have a significant number of lines where our node-selector has zero advantage over the traditional SCIP one, not because our selector is better or worse than SCIP, but simply because it wasn't called in the first place. We set this time-limit relatively low as our prototype selector only runs at the beginning of the solver process, meaning that over time the effects of the traditional solver take over. Running the system for longer yields similar trends, but worse signal-to-noise ratio in the improvement due to the SCIP selector dominating the learnt solver in the long-runtime regime.

A common issue in testing new node selection techniques against an existing (e.g., SCIP) strategy is the degree of code-optimization present in industrial-grade solvers compared to research prototypes: SCIP is a highly optimized C-implementation while our node selector has Python and framework overhead to contend with. This means the node-throughput is naturally going to be much slower than the node-throughput of the baseline, even if we disregard the additional cost of evaluating the neural network. We cannot assess the theoretically possible efficiency of our method, so all of our results should be taken as a lower-bound on performance[4].

### 5.2 EVALUATION METRICS

A core issue in benchmarking is the overall breadth of difficulty and scale of problem instances. Comparing the performance of node selection strategies is challenging due to a lack of aggregatable metrics. Further, the difficulty of the instances in benchmarks do not only depend on the scale but also specific configuration, e.g., distances in TSPLIB: while `swiss42` can be solved quickly, `ulysses22` cannot be solved within our time limit despite only being half the size (see Table 3).

---

[3]Unfortunately, we could not include Labassi et al. (2022) and He et al. (2014) as baselines due to compatibility issues between SCIP versions, see Appendix D for more details.

[4]For instance, our method spends about as much time in the feature-extraction stage as in all other stages combined. This is due to the limited efficiency of even highly optimized Python code.

We can also see this at the range of optimality gaps in Table 3. The gaps range from $1134\%$ to $0\%$. Computing the mean gap alone is not very meaningful as instances with large gaps dominate the average.[5] To facilitate meaningful comparisons, we consider three normalized metrics as follows.

The **Reward** (Eq. 5) considers the shifted *ratio* between the optimality gap of our approach and that of the baseline; positive values represent that our method is better and vice verse. This has the natural advantage of removing the absolute scale of the gaps and only considering relative improvements. The downside is that small differences can get blown-up in cases where the baseline is already small.[6] Note that the function also has an asymmetric range, since one can have an infinitely negative reward, but can have at most have a $+1$ positive reward. Hence, we clip the reward in the range $\pm 1$ as this means a single bad result cannot destroy the entire valuation for either method.

**Utility** defines the *difference* between both methods normalized using the maximum of both gaps:

$$\text{Utility} = \left( \frac{gap(scip) - gap(node\ selector)}{\max\left(gap(node\ selector), gap(scip)\right)} \right). \tag{12}$$

The reason we do not use this as a reward measure is because we empirically found it to produce worse models. This is presumably because some of the negative attributes of our reward, e.g., the asymmetry of the reward signal, lead to more robust policies. In addition, the utility metric gives erroneous measurements when both models hit zero optimality gap. This is because utility implicitly defines $\frac{0}{0} = 0$, rather than reward, which defines it as $\frac{0}{0} = 1$. In some sense the utility measurement is accurate, in that our method does not improve upon the baseline. On the other hand, our method is already provably optimal as soon as it reaches a gap of $0\%$. In general, utility compresses the differences more than reward which may or may not be beneficial in practice.

**Utility per Node** normalizes Utility by the number of nodes used during exploration:

$$\text{Utility/Node} = \left( \frac{scip - selector}{\max\left(selector, scip\right)} \right), \tag{13a}$$

where $selector = \frac{gap(node\ selector)}{nodes(node\ selector)}$ and $scip = \frac{gap(scip)}{nodes(scip)}$. The per-node utility gives a proxy for the total amount of "work" done by each method. However, it ignores the individual node costs, as solving the different LPs may take different amounts of resources (a model with higher "utility/node" is not necessarily more efficient as our learner might pick cheap but lower expected utility nodes on purpose). Further, the metric is underdefined: comparing two ratios, a method may become better by increasing the number of nodes processed, but keeping the achieved gap constant. In practice the number of nodes processed by our node selector is dominated by the implementation rather than the node choices, meaning we can assume it is invariant to changes in policy. Another downside arises if both methods reach zero optimality gap, the resulting efficiency will also be zero regardless of how many nodes we processed. As our method tend to reach optimality much faster (see Sec. 5 and Appendix D), all utility/node results can be seen as a lower-bound for the actual efficiency.

### 5.3 RESULTS

While all results can be found in Appendix D we report an aggregated view for each benchmark in Table 6. In addition to our metrics we report the winning ratio of our method over the baseline, and the geometric mean of the gaps at the end of solving (lower is better).

For benchmarking and training, we leave all settings, such as presolvers, primal heuristics, diving heuristics, constraint specialisations, etc. at their default settings to allow the baseline to perform best. All instances are solved using the same model without any fine-tuning. We expect that tuning, e.g., the aggressiveness of primal heuristics, increases the performance of our method, as it decreases

---

[5]A gap decrease from $1,000\%$ down to $999\%$ has the same overall magnitude as a decrease from $1\%$ to $0\%$ – but from a practical point of view the latter is much more meaningful. The degree of which a result can be improved also depends wildely on the problem's pre-existing optimality gap. For instance an improvement of $2\%$ from $1,000\%$ down to $998\%$ is easily possible, while becoming impossible for a problem whose baseline already achieves only $1\%$ gap. This would mean that in a simple average the small-gap problems would completely vanish under the size of large-gap instances.

[6]For example, if the baseline has a gap of $0.001$ and ours has a gap of $0.002$ our method would be $100\%$ worse, despite the fact that from a practical point of view both of them are almost identical.

Table 1: Performance across benchmarks (the policy only saw TSP instances during training). The 5min runs use the same model, evaluated for the first 650 nodes, and processed according to Sec. 5.1.

| Benchmark | Reward | Utility | Utility/Node | Win-rate | geo-mean Ours | geo-mean SCIP |
|---|---|---|---|---|---|---|
| TSPLIB (Reinelt, 1991) | 0.184 | 0.030 | 0.193 | 0.50 | 0.931 | 0.957 |
| UFLP (Kochetov & Ivanenko, 2005) | 0.078 | 0.093 | -0.064 | 0.636 | 0.491 | 0.520 |
| MINLPLib (Bussieck et al., 2003) | 0.487 | 0.000 | 0.114 | 0.852 | 28.783 | 31.185 |
| MIPLIB (Gleixner et al., 2021) | 0.140 | -0.013 | 0.208 | 0.769 | 545.879 | 848.628 |
| TSPLIB@5min | 0.192 | 0.056 | 0.336 | 0.600 | 1.615 | 2.000 |
| MINLPlib@5min | 0.486 | -0.012 | 0.078 | 0.840 | 17.409 | 20.460 |
| MINLPlib@5min | 0.150 | -0.075 | 0.113 | 0.671 | 66.861 | 106.400 |

the relative cost of evaluating a neural network, but for the sake of comparison we use the same parameters. We train our node selection policy on problem instances according to Sec. 4.3 and apply it on problems from different benchmarks.

First, we will discuss TSPLIB itself, which while dramatically more complex than our selected training instances, still contains instances from the same problem family as the training set (Sec. 5.3.1). Second, we consider instances of the Uncapacitated Facility Location Problem (UFLP) as generated by Kochetov & Ivanenko (2005)'s problem generator. These problems are designed to be particulary challenging to branch-and-bound solvers due to their large optimality gap (Sec. D.3.1). While the first two benchmarks focused on specific problems (giving you a notion of how well the algorithm does on the problem itself) we next consider "'meta-benchmarks" that consist of many different problems, but relatively few instances of each. MINLPLIB (Bussieck et al., 2003) is a meta-benchmark for *nonlinear* mixed-integer programming (Sec. 5.3.2), and MIPLIB (Gleixner et al., 2021) a benchmark for mixed integer programming (Sec. 5.3.3). We also consider generalisation against the uncapacitated facility location problem using a strong instance generator, see Appendix D.3.1. Our benchmarks are diverse and complex and allow to compare algorithmic improvements in state-of-the-art solvers.

### 5.3.1 TSPLIB

From an aggregative viewpoint we outperform the SCIP node selection by $\approx 20\%$ in both reward and utility per node. Due to the scoring of zero-gap instances we are only $3.3\%$ ahead in utility. If both our method and the baseline reach an optimality gap of 0, it is unclear how the normalised reward should appear. "Reward" defines $\frac{0}{0} = 1$ as our method achieved the mathematically optimal value, so it should achieve the optimal reward. "Utility" defines $\frac{0}{0} = 0$ as our method did not improve upon the baseline. While this also persists in "utility per node", our method is much more efficient compared to the baseline s.t. zero-gap problems do not affect our results much.

Qualitatively, it is particularly interesting to study the problems our method still looses against SCIP (in four cases). A possible reason why our method significantly underperforms on `Dantzig42` is that our implementation is just too slow, considering that the baseline manages to evaluate $\approx 40\%$ more nodes. A similar observation can be made on `eil51` where the baseline manages to complete $5\times$ more nodes. `KroE100` is the first instance our method looses against SCIP, although it explores an equal amount of nodes. We believe that this is because our method commits to the wrong subtree early and never manages to correct into the proper subtree. `rd100` is also similar to `Dantzig` and `eil51` as the baseline is able to explore $60\%$ more nodes. Ignoring these four failure cases, our method is either on par (up to stochasticity of the algorithm) or exceeds the baseline significantly.

It is also worthwhile to study the cases where both the baseline and our method hit 0 optimality gap. A quick glance at instances like `bayg29`, `fri26`, `swiss42` or `ulysses16` shows that our method tends to finish these problems with significantly fewer nodes explored. This is not captured by any of our metrics as the "utility/node" metric is zero if the utility is zero, as is the case with 0 optimality gap instances. Qualitatively, instances like `bayg29` manage to reach the optimum in only $\frac{1}{3}$ the number of explored nodes, which showcases a significant improvement in node-selection quality. It is worth noting that, due to the different optimization costs for different nodes, it not always holds that evaluating fewer nodes is faster in wall-clock time. In practice, "fewer nodes is better" seems to be a good rule-of-thumb to check algorithmic efficiency.

### 5.3.2  MINLPLIB

We now consider MINLPs. To solve these, SCIP and other solvers use branching techniques that cut nonlinear (often convex) problems from a relaxed master-relaxation towards true solutions. We consider MINLPLib (Bussieck et al., 2003), a meta-benchmark consisting of hundreds of synthetic and real-world MINLP instances of varying different types and sizes. As some instances take hours to solve (making them inadequate to judge our node selector which mainly aims to improve the starting condition of problems), we also pre-filter the instances. Specifically, we apply the same filtering for tractable problems as in the TSPLIB case. Full results can be found in Appendix D.3.

Our method still manages to outperform SCIP, even on MINLPs, although it has never seen a single MINLP problem before, see Table 6. The reason for the significant divergence between the Reward and Utility performance measures is once again due to the handling of $\frac{0}{0}$. Since MINLPLIB contains a fair few "easy" problems that can be solved to $0\%$ gap, this has a much bigger effect on this benchmark than the others. Qualitatively, our method either outperforms or is on par with the majority of problems, but also loses significantly in some problems, greatly decreasing the average. Despite the fact that utility "rounds down" advantages to zero, the overall utility per node is still significantly better than that of SCIP. Inspecting the instances with poor results, we also see that for most of them the baseline manages to complete significantly more nodes than our underoptimized implementation. We expect features specifically tuned for nonlinear problems to increase performance by additional percentage points, but as feature selection is orthogonal to the actual algorithm design, we leave more thorough discussion of this to future work [7].

### 5.3.3  MIPLIB

Last, but not least we consider the meta-benchmark MIPLIB (Gleixner et al., 2021), which consists of hundreds of real-world mixed-integer programming problems of varying size, complexity, and hardness. Our method is either close to or exceeds the performance of SCIP, see Table 6. It is also the first benchmark our method looses on, according to the utility-metric.

Considering per-instance results, we see similar patterns as in previous failure cases: Often we underperform on instances that need to close many nodes, as our method's throughput lacks behind that of SCIP. We expect that a more efficient implementation alleviates the issues in those cases.

We also see challenges in problems that are far from the training distribution. Consider `fhnw-binpack4-48`, were the baseline yields an optimality gap of $0$ while we end at $+\infty$. This is due to the design of the problem: Instead of a classical optimization problem, this is a satisfaction problem, where not an optimal value, but *any* valid value is searched, i.e., we either yield a gap of $0$, or a gap of $+\infty$, as no other gap is possible. Notably, these kinds of problems may pose a challenge for our algorithm, as the node-pruning dynamics of satisfying MIPs are different than the one for optimizing MIPs: Satisfying MIPs can only rarely prune nodes since, by definition, no intermediary primally valid solutions are ever found. We believe this problem could be solved by considering such problems during training, which we currently do not.

## 6  CONCLUSION

We have proposed a novel approach to branch-and-bound node selection, leveraging the power of bisimulation and RL. By aligning our model with the branch-and-bound tree structure, we have demonstrated the potential to develop a versatile heuristic that can be applied across various optimization problem domains, despite being trained on a narrow set of instances. To our knowledge, this is the first demonstration of learned node selection to mixed-integer (nonlinear) programming.

There are still open questions. Feature selection remains an area where we expect significant improvements, especially for nonlinear programming, which contemporary methods do not account for. We also expect significant improvements in performance through code optimization. An important area for research lies in generalized instance generation: Instead of focusing on single domain instances for training (e.g. from TSP), an instance generator should create problem instances with consistent, but varying levels of difficulty across different problem domains.

---

[7]We are not aware of a learned BnB node-selection heuristic used for MINLPs, so guidance towards feature selection doesn't exist yet. Taking advantage of them presumably also requires to train on nonlinear problems.

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

## A   TSP-AS-MILP FORMULATION

In general, due to the fact that TSP is amongst the most studied problems in discrete optimization, we can expect existing mixed-integer programming systems to have rich heuristics that provide a strong baseline for our method. Mathematically, we choose the Miller–Tucker–Zemlin (MTZ) formulation (Miller et al., 1960):

$$\min_x \quad \sum_{i=1}^{n} \sum_{j \neq i, j=1}^{n} c_{ij} x_{ij} \tag{14a}$$

$$\text{subject to} \quad \sum_{j=1, i \neq j}^{n} x_{ij} = 1 \qquad \forall i = 1, \dots, n \tag{14b}$$

$$\sum_{i=1, i \neq j}^{n} x_{ij} = 1 \qquad \forall j = 1, \dots, n \tag{14c}$$

$$u_1 - u_j + (n-1)x_{ij} \leq n-2 \quad 2 \leq i \neq j \leq n \tag{14d}$$

$$2 \leq u_i \leq n \qquad 2 \leq i \leq n \tag{14e}$$

$$u_i \in \mathbb{Z}, x_{ij} \in \{0,1\} \tag{14f}$$

Effectively this formulation keeps two buffers: one being the actual $(i, j)$-edges travelled $x_{ij}$, the other being a node-order variable $u_i$ that makes sure that $u_i < u_j$ if $i$ is visited before $j$. There are alternative formulations, such as the Dantzig–Fulkerson–Johnson (DFJ) formulation, which are used in modern purpose-built TSP solvers, but those are less useful for general problem generation: The MTZ formulation essentially relaxes the edge-assignments and order constraints, which then are branch-and-bounded into hard assignments during the solving process. This is different to DFJ, which instead relaxes the "has to pass through all nodes" constraint. DFJ allows for subtours (e. g., only contain node $A, B, C$ but not $D, E$) which then get slowly eliminated via the on-the-fly generation of additional constraints. To generate these constraints one needs specialised row-generators which, while very powerful from an optimization point-of-view, make the algorithm less general as a custom row-generator has to intervene into every single node. However, in our usecase we also do not really care about the ultimate performance of individual algorithms as the reinforcement learner only looks for improvements to the existing node selections. This means that as long as the degree of improvement can be adequately judged, we do not need state-of-the-art solver implementations to give the learner a meaningful improvement signal.

## B   UNCAPACITATED FACILITY LOCATION PROBLEM

Mathmatically, the uncapacitated facility location problem can be seen as sending a product $z_{ij}$ from facility $i$ to consumer $j$ with cost $c_{ij}$ and demand $d_j$. One can only send from $i$ to $j$ if facility $i$ was built in the first place, which incurs cost $f_i$. The overall problem therefore is

$$\min_x \quad \sum_{i=1}^{n} \sum_{i=1}^{m} c_{ij} d_j z_{ij} + \sum_{i=0}^{n} f_i x_i \tag{15a}$$

$$\text{subject to} \quad \sum_{j=1, i \neq j}^{n} z_{ij} = 1 \qquad \forall i = 1, \dots, m \tag{15b}$$

$$\sum_{i=1, i \neq j}^{n} z_{ij} \leq M x_i \qquad \forall j = 1, \dots, n \tag{15c}$$

$$z_{ij} \in \{0,1\} \qquad \forall i = 1, \dots, n, \forall j = 1, \dots, m \tag{15d}$$

$$x_i \in \{0,1\} \qquad \forall i = 1, \dots, n \tag{15e}$$

$$\tag{15f}$$

where $M$ is a suitably large constant representing the infinite-capacity one has when constructing $x_i = 1$. One can always choose $M \geq m$ since that, for the purposes of the polytop is equivalent to setting $M$ to literal infinity. This is also sometimes referred to as the "big $M$" method.

The instance generator by Kochetov & Ivanenko (2005) works by setting $n = m = 100$ and setting all opening costs at 3000. Every city has 10 "cheap" connections sampled from $\{0, 1, 2, 3, 4\}$ and the rest have cost 3000, which represents infinity (i. e., also invoking the big $M$ method).

## C  FEATURES

Table 2 lists the features used on every individual node. The features split into two different types: One being "model" features, the other being "node" features. Model features describe the state of the entire model at the currently explored node, while node features are specific to the yet-to-be-solved added node. We aim to normalize all features with respect to problem size, as e. g., just giving the lower-bound to a problem is prone to numerical domain shifts. For instance a problem with objective $c^T x, x \in P$ is inherently the same from a solver point-of-view as a problem $10c^T x, x \in P$, but would give different lower-bounds. Since NNs are generally nonlinear estimators, we need to make sure such changes do not induce huge distribution shifts. We also clamp the feature values between $[-10, 10]$ which represent "infinite" values, which can occur, for example in the optimality gap. Last but not least, we standardize features using empirical mean and standard deviation. These features

Table 2: Features used per individual node.

| | | |
|---|---|---|
| model features | Number of cuts applied | normalized by total number of constraints |
| | Number of separation rounds | |
| | optimality gap | |
| | lp iterations | |
| | mean integrality gap | |
| | percentage of variables already integral | |
| | histogram of fractional part of variables | 10 evenly sized buckets |
| node features | depth of node | normalized by total number of nodes |
| | node lowerbound | normalized by min of primal and dual bound |
| | node estimate | normalized by min of primal and dual bound |

are inspired by prior work, such as Labassi et al. (2022); Yilmaz & Yorke-Smith (2021), but adapted to the fact that we do not need e. g., explicit entries for the left or right child's optimality gap, as these (and more general K-step versions of these) can be handled by the GNN.

Further, to make batching tractable, we aim to have constant size features. This is different from e. g., Labassi et al. (2022), who utilize flexibly sized graphs to represent each node. The upside of this approach is that certain connections between variables and constraints may become more apparent, with the downside being the increased complexity of batching these structures and large amounts of nodes used. This isn't a problem for them, as they only consider pairwise comparisons between nodes, rather than the entire branch-and-bound graph, but for us would induce a great deal of complexity and computational overhead, especially in the larger instances. For this reason, we represent flexibly sized inputs, such as the values of variables, as histograms: i.e., instead of having $k$ nodes for $k$ variables and wiring them together, we produce once distribution of variable values with 10-buckets, and feed this into the network. This looses a bit of detail in the representation, but allows us to scale to much larger instances than ordinarily possible.

In general, these features are not optimized, and we would expect significant improvements from more well-tuned features. Extracting generally meaningful features from branch-and-bound is a nontrivial task and is left as a task for future work.

## D  FULL RESULTS

The following two sections contain the per-instance results on the two "named" benchmarks TSPLIB (Reinelt, 1991) and MINLPLIB (Bussieck et al., 2003). We test against the strong SCIP 8.0.4 baseline. Due to compatibility issues, we decided not to test against (Labassi et al., 2022) or (He et al., 2014): These methods were trained against older versions of SCIP, which not only made running them challenging, but also would not give valid comparisons as we cannot properly account for changes between SCIP versions. Labassi et al. (2022) specifically relies on changes to the SCIP interface, which makes porting to SCIP 8.0.4 intractable. In general, this shouldn't matter too much,

as SCIP is still demonstrably the state-of-the-art non-commercial mixed-integer solver, which frequently outperforms even closed-source commercial solvers (see Mittelmann (2021) for thorough benchmarks against other solvers), meaning outperforming SCIP can be seen as outperforming the state-of-the-art.

## D.1 TSPLIB RESULTS

Table 3: Results on TSPLIB (Reinelt, 1991) after 45s runtime. Note that we filter out problems in which less than 5 nodes were explored as those problems cannot gain meaningful advantages even with perfect node selection. "Name" refers to the instances name, "Gap Base/Ours" corresponds to the optimization gap achieved by the baseline and our method respectively (lower is better), "Nodes Base/Ours" to the number of explored Nodes by each method, and "Reward", "Utility" and "Utility Node" to the different performance measures as described in Section 5.

| Name | Gap Ours | Gap Base | Nodes Ours | Nodes Base | Reward | Utility | Utility/Node |
|------|----------|----------|------------|------------|--------|---------|--------------|
| att48 | 0.287 | 0.286 | 1086 | 2670 | -0.002 | -0.002 | 0.593 |
| bayg29 | 0.000 | 0.000 | 2317 | 7201 | 1.000 | 0.000 | 0.000 |
| bays29 | 0.000 | 0.036 | 11351 | 10150 | 1.000 | 1.000 | 0.997 |
| berlin52 | 0.000 | 0.000 | 777 | 1634 | 1.000 | 0.000 | 0.000 |
| bier127 | 2.795 | 2.777 | 23 | 25 | -0.007 | -0.007 | 0.074 |
| brazil58 | 0.328 | 0.644 | 1432 | 2182 | 0.491 | 0.491 | 0.666 |
| burma14 | 0.000 | 0.000 | 96 | 65 | 1.000 | 0.000 | 0.000 |
| ch130 | 8.801 | 8.783 | 48 | 43 | -0.002 | -0.002 | -0.106 |
| ch150 | 7.803 | 7.802 | 18 | 18 | -0.000 | -0.000 | -0.000 |
| d198 | 0.582 | 0.582 | 10 | 11 | -0.000 | -0.000 | 0.091 |
| dantzig42 | 0.185 | 0.100 | 2498 | 3469 | -0.847 | -0.459 | -0.248 |
| eil101 | 2.434 | 2.430 | 31 | 61 | -0.002 | -0.002 | 0.491 |
| eil51 | 0.178 | 0.017 | 828 | 4306 | -1.000 | -0.907 | -0.514 |
| eil76 | 0.432 | 1.099 | 309 | 709 | 0.607 | 0.607 | 0.829 |
| fri26 | 0.000 | 0.000 | 1470 | 6721 | 1.000 | 0.000 | 0.000 |
| gr120 | 7.078 | 7.083 | 41 | 43 | 0.001 | 0.001 | 0.047 |
| gr137 | 0.606 | 0.603 | 30 | 25 | -0.006 | -0.006 | -0.171 |
| gr17 | 0.000 | 0.000 | 92 | 123 | 1.000 | 0.000 | 0.000 |
| gr24 | 0.000 | 0.000 | 110 | 207 | 1.000 | 0.000 | 0.000 |
| gr48 | 0.192 | 0.340 | 586 | 2479 | 0.435 | 0.435 | 0.866 |
| gr96 | 0.569 | 0.552 | 93 | 182 | -0.032 | -0.031 | 0.472 |
| hk48 | 0.071 | 0.106 | 2571 | 2990 | 0.324 | 0.324 | 0.419 |
| kroA100 | 8.937 | 8.945 | 102 | 233 | 0.001 | 0.001 | 0.563 |
| kroA150 | 11.343 | 11.340 | 23 | 21 | -0.000 | -0.000 | -0.087 |
| kroA200 | 13.726 | 13.723 | 5 | 7 | -0.000 | -0.000 | 0.286 |
| kroB100 | 7.164 | 7.082 | 83 | 109 | -0.011 | -0.011 | 0.230 |
| kroB150 | 10.965 | 10.965 | 16 | 14 | 0.000 | 0.000 | -0.125 |
| kroB200 | 11.740 | 11.740 | 7 | 6 | 0.000 | 0.000 | -0.143 |
| kroC100 | 8.721 | 8.754 | 118 | 133 | 0.004 | 0.004 | 0.116 |
| kroD100 | 7.959 | 7.938 | 70 | 111 | -0.003 | -0.003 | 0.368 |
| kroE100 | 8.573 | 2.952 | 105 | 108 | -1.000 | -0.656 | -0.646 |
| lin105 | 2.005 | 2.003 | 98 | 149 | -0.001 | -0.001 | 0.341 |
| pr107 | 1.367 | 1.336 | 128 | 217 | -0.024 | -0.023 | 0.396 |
| pr124 | 0.937 | 0.935 | 64 | 61 | -0.001 | -0.001 | -0.048 |
| pr136 | 2.351 | 2.350 | 31 | 45 | -0.000 | -0.000 | 0.311 |
| pr144 | 2.228 | 2.200 | 47 | 37 | -0.012 | -0.012 | -0.222 |
| pr152 | 2.688 | 2.683 | 14 | 41 | -0.002 | -0.002 | 0.658 |
| pr226 | 1.091 | 1.092 | 6 | 6 | 0.001 | 0.001 | 0.001 |
| pr76 | 0.534 | 0.476 | 201 | 855 | -0.123 | -0.109 | 0.736 |
| rat99 | 0.853 | 0.849 | 41 | 80 | -0.005 | -0.005 | 0.485 |
| rd100 | 5.948 | 4.462 | 100 | 166 | -0.333 | -0.250 | 0.197 |
| si175 | 0.270 | 0.270 | 8 | 7 | 0.000 | 0.000 | -0.125 |
| st70 | 0.586 | 3.018 | 379 | 1068 | 0.806 | 0.806 | 0.931 |
| swiss42 | 0.000 | 0.000 | 1075 | 1133 | 1.000 | 0.000 | 0.000 |
| ulysses16 | 0.000 | 0.000 | 18322 | 19553 | 1.000 | 0.000 | 0.000 |
| ulysses22 | 0.103 | 0.127 | 13911 | 13313 | 0.191 | 0.191 | 0.154 |
| Mean | — | — | 1321 | 1799 | **0.184** | **0.030** | **0.193** |

## D.2 MIPLIB RESULTS

Table 4: Results on MIPLIB (Gleixner et al., 2021) after 45s runtime. Note that we filter out problems in which less than 5 nodes were explored as those problems cannot gain meaningful advantages even with perfect node selection. "Name" refers to the instances name, "Gap Base/Ours" corresponds to the optimization gap achieved by the baseline and our method respectively (lower is better), "Nodes Base/Ours" to the number of explored Nodes by each method, and "Reward", "Utility" and "Utility Node" to the different performance measures as described in Section 5. Note that all results where achieved with a policy only trained on TSP instances

| Name | Gap Ours | Gap Base | Nodes Ours | Nodes Base | Reward | Utility | Utility/Node |
|---|---|---|---|---|---|---|---|
| 30n20b8 | 2.662 | ∞ | 147 | 301 | 1.000 | 1.000 | 1.000 |
| 50v-10 | 0.101 | 0.113 | 303 | 1094 | 0.103 | 0.103 | 0.752 |
| CMS750_4 | 0.100 | 0.072 | 68 | 281 | -0.389 | -0.280 | 0.664 |
| air05 | 0.000 | 0.000 | 248 | 523 | 1.000 | 0.000 | 0.000 |
| assign1-5-8 | 0.085 | 0.087 | 17466 | 23589 | 0.030 | 0.030 | 0.282 |
| binkar10_1 | 0.000 | 0.000 | 2843 | 2270 | 1.000 | 0.000 | 0.000 |
| blp-ic98 | 0.127 | 0.127 | 26 | 43 | 0.001 | 0.001 | 0.396 |
| bnatt400 | ∞ | ∞ | 547 | 1568 | 0.000 | 0.000 | 0.651 |
| bnatt500 | ∞ | ∞ | 148 | 936 | 0.000 | 0.000 | 0.842 |
| bppc4-08 | 0.038 | 0.038 | 1318 | 3277 | 0.000 | 0.000 | 0.598 |
| cost266-UUE | 0.130 | 0.143 | 468 | 770 | 0.094 | 0.094 | 0.449 |
| csched007 | ∞ | ∞ | 558 | 1770 | 0.000 | 0.000 | 0.685 |
| csched008 | 0.070 | ∞ | 910 | 1179 | 1.000 | 1.000 | 1.000 |
| cvs16r128-89 | 0.560 | 0.601 | 6 | 7 | 0.068 | 0.068 | 0.202 |
| drayage-25-23 | 0.000 | 0.000 | 105 | 267 | 1.000 | 0.000 | 0.000 |
| dws008-01 | ∞ | ∞ | 123 | 173 | 0.000 | 0.000 | 0.289 |
| eil33-2 | 0.194 | 0.189 | 191 | 171 | -0.025 | -0.024 | -0.127 |
| fast0507 | 0.027 | 0.027 | 11 | 7 | -0.003 | -0.003 | -0.366 |
| fastxgemm-n2r6s0t2 | 18.519 | 18.519 | 785 | 2531 | 0.000 | 0.000 | 0.690 |
| fhnw-binpack4-4 | ∞ | ∞ | 140002 | 152608 | 0.000 | 0.000 | 0.083 |
| fhnw-binpack4-48 | ∞ | 0.000 | 15019 | 24649 | -1.000 | -1.000 | -1.000 |
| fiball | 0.029 | 0.036 | 442 | 610 | 0.200 | 0.200 | 0.420 |
| gen-ip002 | 0.008 | 0.010 | 88794 | 125319 | 0.197 | 0.197 | 0.397 |
| gen-ip054 | 0.008 | 0.010 | 157950 | 179874 | 0.207 | 0.207 | 0.263 |
| glass-sc | 0.580 | 0.495 | 200 | 328 | -0.173 | -0.148 | 0.285 |
| glass4 | 1.123 | 1.033 | 37424 | 35671 | -0.087 | -0.080 | -0.123 |
| gmu-35-40 | 0.001 | 0.001 | 28534 | 27077 | 0.402 | 0.398 | 0.276 |
| gmu-35-50 | 0.001 | 0.001 | 16456 | 22333 | 0.177 | 0.176 | 0.346 |
| graph20-20-1rand | 0.000 | 0.000 | 416 | 283 | 1.000 | 0.000 | 0.000 |
| graphdraw-domain | 0.421 | 0.430 | 49640 | 56798 | 0.022 | 0.022 | 0.145 |
| ic97_potential | 0.023 | 0.040 | 39316 | 30633 | 0.415 | 0.415 | 0.247 |
| icir97_tension | 0.011 | 0.006 | 6697 | 7943 | -0.882 | -0.468 | -0.367 |
| irp | 0.000 | 0.000 | 6 | 6 | 1.000 | 0.000 | 0.000 |
| istanbul-no-cutoff | 0.514 | 0.393 | 37 | 28 | -0.309 | -0.236 | -0.422 |
| lectsched-5-obj | ∞ | 2.200 | 1192 | 1118 | -1.000 | -1.000 | -1.000 |
| leo1 | 0.118 | 0.113 | 34 | 108 | -0.049 | -0.046 | 0.670 |
| leo2 | 0.345 | 0.135 | 49 | 61 | -1.000 | -0.609 | -0.514 |
| mad | ∞ | ∞ | 78783 | 81277 | 0.000 | 0.000 | 0.031 |
| markshare2 | ∞ | ∞ | 91135 | 127265 | 0.000 | 0.000 | 0.284 |
| markshare_4_0 | ∞ | ∞ | 570277 | 682069 | 0.000 | 0.000 | 0.164 |
| mas74 | 0.079 | 0.084 | 32005 | 26180 | 0.060 | 0.060 | -0.129 |
| mas76 | 0.014 | 0.015 | 49987 | 52401 | 0.060 | 0.060 | 0.100 |
| mc11 | 0.008 | 0.009 | 333 | 1989 | 0.139 | 0.138 | 0.855 |
| mcsched | 0.090 | 0.086 | 439 | 1526 | -0.049 | -0.046 | 0.698 |
| mik-250-20-75-4 | 0.000 | 0.000 | 10067 | 10120 | 1.000 | 0.000 | 0.000 |
| milo-v12-6-r2-40-1 | 0.038 | 0.031 | 340 | 514 | -0.242 | -0.195 | 0.179 |
| momentum1 | 2.868 | 2.868 | 10 | 9 | -0.000 | -0.000 | -0.100 |
| n2seq36q | 0.665 | 0.665 | 5 | 6 | 0.000 | 0.000 | 0.167 |
| n5-3 | 0.046 | 0.000 | 427 | 595 | -1.000 | -1.000 | -1.000 |
| neos-1171737 | 0.032 | 0.032 | 7 | 13 | 0.000 | 0.000 | 0.462 |
| neos-1445765 | 0.000 | 0.000 | 190 | 263 | 1.000 | 0.000 | 0.000 |
| neos-1456979 | ∞ | 0.344 | 204 | 405 | -1.000 | -1.000 | -1.000 |
| neos-1582420 | 0.016 | 0.016 | 11 | 11 | 0.000 | 0.000 | 0.000 |
| neos-2657525-crna | ∞ | ∞ | 42826 | 45188 | 0.000 | 0.000 | 0.052 |
| neos-2978193-inde | 0.013 | 0.013 | 964 | 2178 | 0.000 | 0.000 | 0.557 |
| neos-3004026-krka | ∞ | ∞ | 1134 | 1163 | 0.000 | 0.000 | 0.025 |
| neos-3024952-loue | ∞ | ∞ | 246 | 377 | 0.000 | 0.000 | 0.347 |
| neos-3046615-murg | 2.515 | 2.631 | 66921 | 79117 | 0.044 | 0.044 | 0.191 |
| neos-3083819-nubu | 0.000 | 0.000 | 1683 | 1687 | 1.000 | 0.000 | 0.000 |
| neos-3381206-awhea | 0.000 | 0.000 | 969 | 230 | 1.000 | 0.000 | 0.000 |
| neos-3402294-bobin | ∞ | ∞ | 10 | 24 | 0.000 | 0.000 | 0.583 |
| neos-3627168-kasai | 0.003 | 0.008 | 6269 | 3338 | 0.577 | 0.577 | 0.205 |
| neos-3754480-nidda | ∞ | ∞ | 87703 | 106632 | 0.000 | 0.000 | 0.178 |

| Name | Gap Ours | Gap Base | Nodes Ours | Nodes Base | Reward | Utility | Utility/Node |
|---|---|---|---|---|---|---|---|
| neos-4338804-snowy | 0.024 | 0.028 | 37447 | 36741 | 0.125 | 0.125 | 0.107 |
| neos-4387871-tavua | 0.631 | 0.634 | 5 | 7 | 0.005 | 0.005 | 0.289 |
| neos-4738912-atrato | 0.016 | 0.006 | 529 | 1064 | -1.000 | -0.634 | -0.265 |
| neos-4954672-berkel | 0.265 | 0.254 | 454 | 775 | -0.043 | -0.041 | 0.389 |
| neos-5093327-huahum | 0.539 | 0.559 | 5 | 6 | 0.036 | 0.036 | 0.197 |
| neos-5107597-kakapo | 2.639 | 5.077 | 1885 | 2332 | 0.480 | 0.480 | 0.580 |
| neos-5188808-nattai | ∞ | ∞ | 16 | 105 | 0.000 | 0.000 | 0.848 |
| neos-5195221-niemur | 106.417 | 106.417 | 11 | 12 | 0.000 | 0.000 | 0.083 |
| neos-911970 | 0.000 | 0.000 | 3905 | 15109 | 1.000 | 0.000 | 0.000 |
| neos17 | 0.000 | 0.000 | 2151 | 3346 | 1.000 | 0.000 | 0.000 |
| neos5 | 0.062 | 0.059 | 66231 | 91449 | -0.053 | -0.050 | 0.235 |
| neos859080 | 0.000 | 0.000 | 990 | 1227 | 1.000 | 0.000 | 0.000 |
| net12 | 2.592 | 2.114 | 56 | 29 | -0.227 | -0.185 | -0.578 |
| ns1208400 | ∞ | ∞ | 82 | 150 | 0.000 | 0.000 | 0.453 |
| ns1830653 | 2.831 | 1.242 | 334 | 686 | -1.000 | -0.561 | -0.099 |
| ns1952667 | ∞ | ∞ | 100 | 52 | 0.000 | 0.000 | -0.480 |
| nu25-pr12 | 0.000 | 0.000 | 119 | 153 | 1.000 | 0.000 | 0.000 |
| nursesched-sprint02 | 0.000 | 0.000 | 9 | 7 | 1.000 | 0.000 | 0.000 |
| nw04 | 0.000 | 0.000 | 6 | 6 | 1.000 | 0.000 | 0.000 |
| pg | 0.000 | 0.000 | 460 | 491 | 1.000 | 0.000 | 0.000 |
| pg5_34 | 0.004 | 0.004 | 275 | 592 | -0.023 | -0.022 | 0.524 |
| piperout-08 | 0.000 | 0.000 | 223 | 309 | 1.000 | 0.000 | 0.000 |
| piperout-27 | 0.000 | 0.000 | 47 | 28 | 1.000 | 0.000 | 0.000 |
| pk1 | 1.244 | 1.117 | 102268 | 120685 | -0.113 | -0.102 | 0.057 |
| radiationm18-12-05 | 0.057 | 0.167 | 886 | 2569 | 0.661 | 0.661 | 0.883 |
| rail507 | 0.033 | 0.033 | 10 | 9 | 0.000 | 0.000 | -0.100 |
| ran14x18-disj-8 | 0.115 | 0.092 | 458 | 975 | -0.251 | -0.200 | 0.412 |
| rd-rplusc-21 | ∞ | ∞ | 137 | 3542 | 0.000 | 0.000 | 0.961 |
| reblock115 | 0.106 | 0.139 | 80 | 731 | 0.238 | 0.238 | 0.917 |
| rmatr100-p10 | 0.216 | 0.326 | 43 | 74 | 0.337 | 0.337 | 0.615 |
| rocI-4-11 | 0.671 | 0.837 | 12054 | 7909 | 0.198 | 0.198 | -0.181 |
| rocII-5-11 | 3.479 | 1.568 | 164 | 287 | -1.000 | -0.549 | -0.211 |
| rococoB10-011000 | 1.244 | 1.258 | 12 | 26 | 0.012 | 0.012 | 0.544 |
| rococoC10-001000 | 0.337 | 0.153 | 135 | 866 | -1.000 | -0.546 | 0.656 |
| roll3000 | 0.000 | 0.000 | 1156 | 2046 | 1.000 | 0.000 | 0.000 |
| sct2 | 0.001 | 0.002 | 2117 | 1215 | 0.619 | 0.615 | 0.332 |
| seymour | 0.044 | 0.035 | 176 | 563 | -0.243 | -0.195 | 0.611 |
| seymour1 | 0.003 | 0.003 | 329 | 885 | 0.146 | 0.145 | 0.682 |
| sp150x300d | 0.000 | 0.000 | 148 | 124 | 1.000 | 0.000 | 0.000 |
| supportcase18 | 0.081 | 0.081 | 178 | 1372 | 0.000 | -0.000 | 0.870 |
| supportcase26 | 0.224 | 0.231 | 11191 | 20287 | 0.031 | 0.031 | 0.465 |
| supportcase33 | 27.788 | 0.371 | 15 | 28 | -1.000 | -0.987 | -0.975 |
| supportcase40 | 0.086 | 0.094 | 50 | 111 | 0.087 | 0.087 | 0.589 |
| supportcase42 | 0.033 | 0.050 | 76 | 256 | 0.340 | 0.340 | 0.804 |
| swath1 | 0.000 | 0.000 | 311 | 372 | 1.000 | 0.000 | 0.000 |
| swath3 | 0.110 | 0.113 | 1442 | 2800 | 0.020 | 0.020 | 0.495 |
| timtab1 | 0.126 | 0.094 | 22112 | 25367 | -0.333 | -0.250 | -0.139 |
| tr12-30 | 0.002 | 0.002 | 8941 | 14896 | 0.019 | 0.019 | 0.394 |
| traininstance2 | ∞ | ∞ | 412 | 821 | 0.000 | 0.000 | 0.498 |
| traininstance6 | 29.355 | ∞ | 2549 | 6376 | 1.000 | 1.000 | 1.000 |
| trento1 | 3.885 | 3.885 | 4 | 7 | -0.000 | -0.000 | 0.429 |
| uct-subprob | 0.249 | 0.195 | 225 | 263 | -0.276 | -0.216 | -0.084 |
| var-smallemery-m6j6 | 0.062 | 0.062 | 95 | 224 | -0.002 | -0.002 | 0.575 |
| wachplan | 0.125 | 0.125 | 422 | 712 | 0.000 | 0.000 | 0.407 |
| Mean | — | — | 16538 | 19673 | **0.140** | -0.013 | **0.208** |

## D.3 MINLPLIB RESULTS

Table 5: Results on MINLPLIB (Bussieck et al., 2003) after 45s runtime. Note that we filter out problems in which less than 5 nodes were explored as those problems cannot gain meaningful advantages even with perfect node selection. "Name" refers to the instances name, "Gap Base/Ours" corresponds to the optimization gap achieved by the baseline and our method respectively (lower is better), "Nodes Base/Ours" to the number of explored Nodes by each method, and "Reward", "Utility" and "Utility Node" to the different performance measures as described in Section 5. For all three measures, higher is better.

| Name | Gap Ours | Gap Base | Nodes Ours | Nodes Base | Reward | Utility | Utility/Node |
|---|---|---|---|---|---|---|---|
| ball_mk4_05 | 0.000 | 0.000 | 1819 | 1869 | 1.000 | 0.000 | 0.000 |
| ball_mk4_10 | ∞ | ∞ | 31684 | 37656 | 0.000 | 0.000 | 0.159 |

| Name | Gap Ours | Gap Base | Nodes Ours | Nodes Base | Reward | Utility | Utility/Node |
|---|---|---|---|---|---|---|---|
| ball_mk4_15 | ∞ | ∞ | 1773 | 2415 | 0.000 | 0.000 | 0.266 |
| bayes2_20 | ∞ | ∞ | 3171 | 2719 | 0.000 | 0.000 | -0.143 |
| bayes2_30 | ∞ | ∞ | 4462 | 4992 | 0.000 | 0.000 | 0.106 |
| bayes2_50 | ∞ | ∞ | 2934 | 2530 | 0.000 | 0.000 | -0.138 |
| blend029 | 0.000 | 0.000 | 812 | 804 | 1.000 | 0.000 | 0.000 |
| blend146 | 0.097 | 0.105 | 12390 | 18066 | 0.075 | 0.075 | 0.365 |
| blend480 | 0.071 | 0.000 | 4878 | 6312 | -1.000 | -1.000 | -0.999 |
| blend531 | 0.000 | 0.000 | 3150 | 7161 | 1.000 | 0.000 | 0.000 |
| blend718 | 0.898 | 0.796 | 22652 | 26060 | -0.127 | -0.113 | 0.020 |
| blend721 | 0.000 | 0.000 | 4650 | 2708 | 1.000 | 0.000 | 0.000 |
| blend852 | 0.021 | 0.000 | 7726 | 5413 | -1.000 | -1.000 | -0.997 |
| camshape100 | 0.076 | 0.074 | 18839 | 22205 | -0.027 | -0.026 | 0.128 |
| camshape200 | 0.145 | 0.147 | 8199 | 9921 | 0.012 | 0.012 | 0.183 |
| camshape400 | 0.198 | 0.195 | 4324 | 5275 | -0.016 | -0.016 | 0.167 |
| camshape800 | 0.222 | 0.226 | 1504 | 1627 | 0.019 | 0.019 | 0.093 |
| cardqp_inlp | 1.436 | 1.660 | 4316 | 7232 | 0.135 | 0.135 | 0.484 |
| cardqp_iqp | 1.089 | 1.660 | 4766 | 7285 | 0.344 | 0.344 | 0.571 |
| carton7 | 0.000 | 0.000 | 55 | 73 | 1.000 | 0.000 | 0.000 |
| carton9 | 0.000 | 0.000 | 9848 | 7406 | 1.000 | 0.000 | 0.000 |
| catmix100 | ∞ | ∞ | 186 | 8750 | 0.000 | 0.000 | 0.979 |
| catmix200 | ∞ | ∞ | 123 | 3870 | 0.000 | 0.000 | 0.968 |
| catmix400 | ∞ | ∞ | 146 | 3498 | 0.000 | 0.000 | 0.958 |
| catmix800 | ∞ | ∞ | 75 | 333 | 0.000 | 0.000 | 0.775 |
| celar6-sub0 | ∞ | ∞ | 4 | 6 | 0.000 | 0.000 | 0.333 |
| chimera_k64ising-01 | 0.701 | 16.469 | 18 | 21 | 0.957 | 0.957 | 0.964 |
| chimera_k64maxcut-01 | 0.523 | 0.199 | 57 | 198 | -1.000 | -0.618 | 0.246 |
| chimera_k64maxcut-02 | 0.368 | 0.239 | 72 | 381 | -0.536 | -0.349 | 0.710 |
| chimera_lga-02 | 0.893 | 0.893 | 5 | 6 | 0.000 | 0.000 | 0.167 |
| chimera_mgw-c8-439-onc8-001 | 0.045 | 0.021 | 127 | 521 | -1.000 | -0.529 | 0.482 |
| chimera_mgw-c8-439-onc8-002 | 0.067 | 0.046 | 72 | 526 | -0.449 | -0.310 | 0.802 |
| chimera_mgw-c8-507-onc8-01 | 0.232 | 0.233 | 26 | 99 | 0.003 | 0.003 | 0.738 |
| chimera_mgw-c8-507-onc8-02 | 0.188 | 0.346 | 14 | 25 | 0.455 | 0.455 | 0.695 |
| chimera_mis-01 | 0.000 | 0.000 | 7 | 7 | 1.000 | 0.000 | 0.000 |
| chimera_mis-02 | 0.000 | 0.000 | 7 | 7 | 1.000 | 0.000 | 0.000 |
| chimera_rfr-01 | 1.029 | 1.153 | 70 | 61 | 0.108 | 0.108 | -0.023 |
| chimera_rfr-02 | 1.148 | 1.061 | 74 | 63 | -0.082 | -0.076 | -0.213 |
| chimera_selby-c8-onc8-01 | 0.436 | 0.224 | 34 | 111 | -0.941 | -0.485 | 0.406 |
| chimera_selby-c8-onc8-02 | 0.439 | 0.232 | 40 | 92 | -0.895 | -0.472 | 0.176 |
| clay0203m | 0.000 | 0.000 | 19 | 30 | 1.000 | 0.000 | 0.000 |
| clay0204m | 0.000 | 0.000 | 266 | 400 | 1.000 | 0.000 | 0.000 |
| clay0205m | 0.000 | 0.000 | 4058 | 3908 | 1.000 | 0.000 | 0.000 |
| clay0303m | 0.000 | 0.000 | 107 | 45 | 1.000 | 0.000 | 0.000 |
| clay0304m | 0.000 | 0.000 | 337 | 897 | 1.000 | 0.000 | 0.000 |
| clay0305m | 0.000 | 0.000 | 4057 | 4204 | 1.000 | 0.000 | 0.000 |
| color_lab3_3x0 | 1.445 | 1.725 | 320 | 576 | 0.162 | 0.162 | 0.534 |
| color_lab3_4x0 | 5.581 | 5.455 | 265 | 434 | -0.023 | -0.023 | 0.375 |
| crossdock_15x7 | 4.457 | 8.216 | 654 | 1080 | 0.458 | 0.458 | 0.672 |
| crossdock_15x8 | 8.578 | 84.148 | 391 | 717 | 0.898 | 0.898 | 0.944 |
| crudeoil_lee1_06 | 0.000 | 0.000 | 48 | 57 | 1.000 | 0.000 | 0.000 |
| crudeoil_lee1_07 | 0.000 | 0.000 | 57 | 92 | 1.000 | 0.000 | 0.000 |
| crudeoil_lee1_08 | 0.000 | 0.000 | 161 | 121 | 1.000 | 0.000 | 0.000 |
| crudeoil_lee1_09 | 0.000 | 0.000 | 107 | 99 | 1.000 | 0.000 | 0.000 |
| crudeoil_lee1_10 | 0.000 | 0.000 | 78 | 109 | 1.000 | 0.000 | 0.000 |
| crudeoil_lee2_05 | 0.000 | 0.000 | 10 | 11 | 1.000 | 0.000 | 0.000 |
| crudeoil_lee2_06 | 0.000 | 0.000 | 45 | 109 | 1.000 | 0.000 | 0.000 |
| crudeoil_lee2_07 | 0.000 | 0.000 | 286 | 81 | 1.000 | 0.000 | 0.000 |
| crudeoil_lee2_08 | 0.000 | 0.000 | 150 | 308 | 1.000 | 0.000 | 0.000 |
| crudeoil_lee2_09 | 0.142 | 0.015 | 44 | 41 | -1.000 | -0.897 | -0.904 |
| crudeoil_lee3_05 | 0.000 | 0.000 | 1435 | 1820 | 1.000 | 0.000 | 0.000 |
| crudeoil_lee3_06 | 0.057 | 0.013 | 352 | 1349 | -1.000 | -0.764 | -0.095 |
| crudeoil_lee4_05 | 0.000 | 0.000 | 306 | 118 | 1.000 | 0.000 | 0.000 |
| crudeoil_lee4_06 | 0.000 | 0.000 | 129 | 60 | 1.000 | 0.000 | 0.000 |
| crudeoil_lee4_07 | 0.000 | 0.000 | 193 | 89 | 1.000 | 0.000 | 0.000 |
| crudeoil_lee4_08 | 0.000 | 0.001 | 41 | 53 | 0.187 | 0.184 | 0.371 |
| crudeoil_li01 | 0.049 | 0.017 | 16819 | 11797 | -1.000 | -0.657 | -0.758 |
| crudeoil_li02 | 0.013 | 0.013 | 12172 | 10426 | -0.027 | -0.027 | -0.165 |
| crudeoil_li03 | ∞ | ∞ | 198 | 899 | 0.000 | 0.000 | 0.780 |
| crudeoil_li05 | 0.157 | 0.142 | 553 | 1031 | -0.104 | -0.095 | 0.408 |
| crudeoil_li06 | ∞ | ∞ | 41 | 322 | 0.000 | 0.000 | 0.873 |
| crudeoil_li11 | ∞ | ∞ | 20 | 70 | 0.000 | 0.000 | 0.714 |
| crudeoil_pooling_ct1 | 0.943 | 0.988 | 2415 | 6356 | 0.046 | 0.046 | 0.638 |
| crudeoil_pooling_ct2 | 0.000 | 0.000 | 1480 | 1589 | 1.000 | 0.000 | 0.000 |
| crudeoil_pooling_ct3 | 42.222 | 120.618 | 101 | 101 | 0.650 | 0.650 | 0.650 |
| crudeoil_pooling_ct4 | 0.000 | 0.000 | 7631 | 9217 | 0.365 | 0.153 | 0.041 |
| du-opt | 0.000 | 0.000 | 11282 | 14174 | 1.000 | 0.000 | 0.000 |
| du-opt5 | 0.000 | 0.000 | 83 | 60 | 1.000 | 0.000 | 0.000 |
| edgecross10-030 | 0.000 | 0.000 | 7 | 7 | 1.000 | 0.000 | 0.000 |

Continued on next page

| Name | Gap Ours | Gap Base | Nodes Ours | Nodes Base | Reward | Utility | Utility/Node |
|---|---|---|---|---|---|---|---|
| edgecross10-040 | 0.000 | 0.000 | 30 | 39 | 1.000 | 0.000 | 0.000 |
| edgecross10-050 | 0.000 | 0.000 | 487 | 469 | 1.000 | 0.000 | 0.000 |
| edgecross10-060 | 0.000 | 0.000 | 2058 | 2138 | 1.000 | 0.000 | 0.000 |
| edgecross10-070 | 0.321 | 0.220 | 255 | 329 | -0.457 | -0.314 | -0.115 |
| edgecross10-080 | 0.077 | 0.077 | 352 | 668 | 0.001 | 0.001 | 0.474 |
| edgecross10-090 | 0.000 | 0.000 | 7 | 6 | 1.000 | 0.000 | 0.000 |
| edgecross14-039 | 0.000 | 0.000 | 624 | 731 | 1.000 | 0.000 | 0.000 |
| edgecross14-058 | 1.251 | 0.549 | 84 | 157 | -1.000 | -0.561 | -0.180 |
| edgecross14-078 | 1.843 | 1.865 | 12 | 14 | 0.012 | 0.012 | 0.153 |
| edgecross14-098 | 1.120 | 1.129 | 24 | 31 | 0.007 | 0.007 | 0.232 |
| edgecross14-117 | 0.963 | 0.947 | 9 | 17 | -0.017 | -0.017 | 0.462 |
| edgecross14-137 | 0.537 | 0.552 | 20 | 30 | 0.028 | 0.028 | 0.352 |
| edgecross14-156 | 0.338 | 0.353 | 13 | 13 | 0.042 | 0.042 | 0.042 |
| edgecross14-176 | 0.089 | 0.080 | 37 | 135 | -0.117 | -0.105 | 0.694 |
| edgecross20-040 | 0.000 | 0.000 | 71 | 57 | 1.000 | 0.000 | 0.000 |
| edgecross20-080 | 3.943 | 3.943 | 7 | 7 | 0.000 | 0.000 | 0.000 |
| edgecross22-048 | 0.615 | 0.000 | 56 | 81 | -1.000 | -1.000 | -1.000 |
| edgecross24-057 | 5.219 | 5.219 | 7 | 6 | 0.000 | 0.000 | -0.143 |
| elf | 0.000 | 0.000 | 115 | 112 | 1.000 | 0.000 | 0.000 |
| ex2_1_1 | 0.000 | 0.000 | 17 | 17 | 1.000 | 0.000 | 0.000 |
| ex2_1_10 | 0.000 | 0.000 | 13 | 11 | 1.000 | 0.000 | 0.000 |
| ex2_1_5 | 0.000 | 0.000 | 17 | 19 | 1.000 | 0.000 | 0.000 |
| ex2_1_6 | 0.000 | 0.000 | 13 | 13 | 1.000 | 0.000 | 0.000 |
| ex2_1_7 | 0.000 | 0.000 | 1523 | 1831 | 1.000 | 0.000 | 0.000 |
| ex2_1_8 | 0.000 | 0.000 | 75 | 93 | 1.000 | 0.000 | 0.000 |
| ex2_1_9 | 0.000 | 0.000 | 3735 | 3947 | 1.000 | 0.000 | 0.000 |
| ex3_1_1 | 0.000 | 0.000 | 405 | 271 | 1.000 | 0.000 | 0.000 |
| ex3_1_3 | 0.000 | 0.000 | 21 | 27 | 1.000 | 0.000 | 0.000 |
| ex3_1_4 | 0.000 | 0.000 | 23 | 23 | 1.000 | 0.000 | 0.000 |
| ex4 | 0.000 | 0.000 | 23 | 29 | 1.000 | 0.000 | 0.000 |
| ex5_2_2_case1 | 0.000 | 0.000 | 39 | 19 | 1.000 | 0.000 | 0.000 |
| ex5_2_2_case2 | 0.000 | 0.000 | 57 | 31 | 1.000 | 0.000 | 0.000 |
| ex5_2_4 | 0.000 | 0.000 | 251 | 227 | 1.000 | 0.000 | 0.000 |
| ex5_2_5 | 0.359 | 0.346 | 30403 | 33492 | -0.038 | -0.036 | 0.058 |
| ex5_3_2 | 0.000 | 0.000 | 33 | 31 | 1.000 | 0.000 | 0.000 |
| ex5_3_3 | 0.339 | 0.331 | 29464 | 31558 | -0.024 | -0.024 | 0.044 |
| ex5_4_2 | 0.000 | 0.000 | 41 | 35 | 1.000 | 0.000 | 0.000 |
| ex8_3_2 | 23.252 | 23.608 | 8907 | 8680 | 0.015 | 0.015 | -0.011 |
| ex8_3_3 | 23.004 | 23.004 | 9636 | 10365 | 0.000 | 0.000 | 0.070 |
| ex8_3_4 | 1.817 | 1.793 | 9447 | 9563 | -0.013 | -0.013 | -0.001 |
| ex8_3_5 | 143.677 | 143.677 | 9427 | 9699 | 0.000 | 0.000 | 0.028 |
| ex8_3_8 | 2.071 | 2.071 | 2293 | 3677 | 0.000 | 0.000 | 0.376 |
| ex8_3_9 | 12.106 | 12.106 | 14272 | 17310 | 0.000 | -0.000 | 0.176 |
| ex8_4_1 | 0.000 | 0.000 | 670 | 650 | 1.000 | 0.000 | 0.000 |
| ex9_2_3 | 0.000 | 0.000 | 25 | 31 | 1.000 | 0.000 | 0.000 |
| ex9_2_5 | 0.000 | 0.000 | 27 | 29 | 1.000 | 0.000 | 0.000 |
| ex9_2_7 | 0.000 | 0.000 | 11 | 11 | 1.000 | 0.000 | 0.000 |
| faclay20h | 1.727 | 1.727 | 16 | 15 | 0.000 | 0.000 | -0.062 |
| faclay25 | 2.468 | 2.468 | 6 | 6 | 0.000 | 0.000 | 0.000 |
| forest | 0.003 | 0.020 | 29002 | 25913 | 0.860 | 0.859 | 0.831 |
| gabriel01 | 0.139 | 0.139 | 6753 | 9744 | -0.000 | -0.000 | 0.307 |
| gabriel02 | 0.556 | 0.585 | 1107 | 1675 | 0.050 | 0.050 | 0.372 |
| gabriel04 | ∞ | 1.308 | 129 | 285 | -1.000 | -1.000 | -1.000 |
| gabriel05 | ∞ | ∞ | 141 | 326 | 0.000 | 0.000 | 0.567 |
| gasprod_sarawak01 | 0.000 | 0.000 | 11 | 6 | 1.000 | 0.000 | 0.000 |
| gasprod_sarawak16 | 0.004 | 0.009 | 506 | 1052 | 0.585 | 0.585 | 0.800 |
| genpooling_lee1 | 0.000 | 0.000 | 690 | 676 | 1.000 | 0.000 | 0.000 |
| genpooling_lee2 | 0.000 | 0.000 | 1299 | 2989 | 1.000 | 0.000 | 0.000 |
| genpooling_meyer04 | 0.957 | 0.691 | 12855 | 17889 | -0.385 | -0.278 | 0.005 |
| genpooling_meyer10 | 1.276 | 1.385 | 1910 | 2815 | 0.078 | 0.078 | 0.375 |
| genpooling_meyer15 | 6.080 | 0.691 | 97 | 413 | -1.000 | -0.886 | -0.516 |
| graphpart_2g-0099-9211 | 0.000 | 0.000 | 18 | 14 | 1.000 | 0.000 | 0.000 |
| graphpart_2pm-0077-0777 | 0.000 | 0.000 | 5 | 6 | 1.000 | 0.000 | 0.000 |
| graphpart_2pm-0088-0888 | 0.000 | 0.000 | 9 | 7 | 1.000 | 0.000 | 0.000 |
| graphpart_2pm-0099-0999 | 0.000 | 0.000 | 16 | 12 | 1.000 | 0.000 | 0.000 |
| graphpart_3g-0334-0334 | 0.000 | 0.000 | 21 | 41 | 1.000 | 0.000 | 0.000 |
| graphpart_3g-0344-0344 | 0.000 | 0.000 | 61 | 19 | 1.000 | 0.000 | 0.000 |
| graphpart_3g-0444-0444 | 0.000 | 0.000 | 424 | 562 | 1.000 | 0.000 | 0.000 |
| graphpart_3pm-0244-0244 | 0.000 | 0.000 | 21 | 15 | 1.000 | 0.000 | 0.000 |
| graphpart_3pm-0334-0334 | 0.000 | 0.000 | 20 | 38 | 1.000 | 0.000 | 0.000 |
| graphpart_3pm-0344-0344 | 0.000 | 0.000 | 590 | 619 | 1.000 | 0.000 | 0.000 |
| graphpart_3pm-0444-0444 | 0.058 | 0.000 | 755 | 1348 | -1.000 | -1.000 | -1.000 |
| graphpart_clique-20 | 0.000 | 0.000 | 22 | 24 | 1.000 | 0.000 | 0.000 |
| graphpart_clique-30 | 0.000 | 0.000 | 421 | 337 | 1.000 | 0.000 | 0.000 |
| graphpart_clique-40 | 1.018 | 0.920 | 297 | 609 | -0.106 | -0.096 | 0.461 |
| graphpart_clique-50 | 5.638 | 6.032 | 97 | 191 | 0.065 | 0.065 | 0.525 |
| graphpart_clique-60 | 17.434 | 9.335 | 109 | 204 | -0.868 | -0.465 | 0.002 |

| Name | Gap Ours | Gap Base | Nodes Ours | Nodes Base | Reward | Utility | Utility/Node |
|---|---|---|---|---|---|---|---|
| graphpart_clique-70 | 30.409 | 35.053 | 16 | 27 | 0.132 | 0.132 | 0.486 |
| haverly | 0.000 | 0.000 | 45 | 57 | 1.000 | 0.000 | 0.000 |
| himmel16 | 0.000 | 0.000 | 2193 | 2089 | 1.000 | 0.000 | 0.000 |
| house | 0.000 | 0.000 | 58675 | 58399 | 1.000 | 0.000 | 0.000 |
| hvb11 | 0.018 | 0.182 | 19172 | 15631 | 0.899 | 0.899 | 0.875 |
| hydroenergy1 | 0.007 | 0.007 | 15060 | 18149 | -0.088 | -0.081 | 0.095 |
| hydroenergy2 | 0.016 | 0.016 | 4834 | 6712 | 0.038 | 0.038 | 0.306 |
| hydroenergy3 | 0.022 | 0.023 | 565 | 1060 | 0.006 | 0.006 | 0.470 |
| ising2_5-300_5555 | 0.508 | 0.407 | 57 | 220 | -0.248 | -0.199 | 0.677 |
| kall_circles_c6a | 3.180 | 2.094 | 42813 | 46497 | -0.519 | -0.342 | -0.285 |
| kall_circles_c6b | 2.635 | 1.452 | 38722 | 45596 | -0.815 | -0.449 | -0.351 |
| kall_circles_c6c | ∞ | ∞ | 33357 | 36374 | 0.000 | 0.000 | 0.083 |
| kall_circles_c7a | 1.482 | 1.376 | 38682 | 43723 | -0.077 | -0.072 | 0.047 |
| kall_circles_c8a | ∞ | ∞ | 32114 | 36262 | 0.000 | 0.000 | 0.114 |
| kall_circlespolygons_c1p12 | 0.000 | 0.000 | 44439 | 64102 | -1.000 | -0.733 | -0.106 |
| kall_circlespolygons_c1p13 | 0.000 | 0.000 | 8621 | 7914 | 1.000 | 0.000 | 0.000 |
| kall_circlespolygons_c1p5a | ∞ | ∞ | 12369 | 13200 | 0.000 | 0.000 | 0.063 |
| kall_circlespolygons_c1p6a | ∞ | ∞ | 404 | 628 | 0.000 | 0.000 | 0.357 |
| kall_circlesrectangles_c1r12 | 0.000 | 0.000 | 42587 | 48285 | 0.121 | 0.114 | 0.061 |
| kall_circlesrectangles_c1r13 | 0.000 | 0.000 | 4372 | 3739 | 1.000 | 0.000 | 0.000 |
| kall_circlesrectangles_c6r1 | ∞ | ∞ | 5850 | 7908 | 0.000 | 0.000 | 0.260 |
| kall_circlesrectangles_c6r29 | ∞ | ∞ | 4181 | 5220 | 0.000 | 0.000 | 0.199 |
| kall_circlesrectangles_c6r39 | ∞ | ∞ | 2570 | 2966 | 0.000 | 0.000 | 0.134 |
| kall_congruentcircles_c31 | 0.000 | 0.000 | 101 | 95 | 1.000 | 0.000 | 0.000 |
| kall_congruentcircles_c32 | 0.000 | 0.000 | 133 | 139 | 1.000 | 0.000 | 0.000 |
| kall_congruentcircles_c41 | 0.000 | 0.000 | 27 | 31 | 1.000 | 0.000 | 0.000 |
| kall_congruentcircles_c42 | 0.000 | 0.000 | 205 | 125 | 1.000 | 0.000 | 0.000 |
| kall_congruentcircles_c51 | 0.000 | 0.000 | 4197 | 4987 | 1.000 | 0.000 | 0.000 |
| kall_congruentcircles_c52 | 0.000 | 0.000 | 1767 | 1446 | 1.000 | 0.000 | 0.000 |
| kall_congruentcircles_c61 | 0.000 | 0.000 | 27338 | 35199 | 1.000 | 0.000 | 0.000 |
| kall_congruentcircles_c62 | 0.000 | 0.000 | 2879 | 6037 | 1.000 | 0.000 | 0.000 |
| kall_congruentcircles_c63 | 0.000 | 0.000 | 2043 | 1729 | 1.000 | 0.000 | 0.000 |
| kall_congruentcircles_c71 | ∞ | ∞ | 39102 | 43349 | 0.000 | 0.000 | 0.098 |
| kall_congruentcircles_c72 | 0.000 | 0.000 | 14686 | 14089 | 1.000 | 0.000 | 0.000 |
| kall_diffcircles_10 | 2.276 | 4.054 | 32475 | 41241 | 0.439 | 0.439 | 0.558 |
| kall_diffcircles_5a | 0.000 | 0.000 | 2020 | 1218 | 1.000 | 0.000 | 0.000 |
| kall_diffcircles_5b | 0.000 | 0.000 | 6360 | 5774 | 1.000 | 0.000 | 0.000 |
| kall_diffcircles_6 | 0.000 | 0.000 | 2827 | 2383 | 1.000 | 0.000 | 0.000 |
| kall_diffcircles_7 | 0.000 | 0.000 | 9408 | 9518 | 1.000 | 0.000 | 0.000 |
| kall_diffcircles_8 | 0.406 | 0.219 | 48924 | 57747 | -0.851 | -0.460 | -0.362 |
| kall_diffcircles_9 | 1.676 | 1.052 | 42056 | 48915 | -0.594 | -0.373 | -0.270 |
| knp3-12 | 1.846 | 1.963 | 1987 | 2132 | 0.060 | 0.060 | 0.124 |
| lop97ic | ∞ | ∞ | 19 | 33 | 0.000 | 0.000 | 0.424 |
| lop97icx | 0.008 | 0.000 | 3041 | 1711 | -1.000 | -0.999 | -0.998 |
| maxcsp-langford-3-11 | ∞ | ∞ | 1356 | 4038 | 0.000 | 0.000 | 0.664 |
| ndcc12 | ∞ | ∞ | 1394 | 3975 | 0.000 | 0.000 | 0.649 |
| ndcc12persp | ∞ | ∞ | 1092 | 2994 | 0.000 | 0.000 | 0.635 |
| ndcc13 | ∞ | ∞ | 298 | 787 | 0.000 | 0.000 | 0.621 |
| ndcc13persp | 0.536 | 0.546 | 2982 | 5662 | 0.018 | 0.018 | 0.483 |
| ndcc14 | 1.030 | 1.048 | 234 | 499 | 0.018 | 0.018 | 0.539 |
| ndcc14persp | 1.044 | 1.080 | 572 | 1052 | 0.033 | 0.033 | 0.474 |
| ndcc15 | ∞ | ∞ | 1293 | 2120 | 0.000 | 0.000 | 0.390 |
| ndcc15persp | ∞ | ∞ | 5227 | 6549 | 0.000 | 0.000 | 0.202 |
| ndcc16 | ∞ | ∞ | 407 | 396 | 0.000 | 0.000 | -0.027 |
| ndcc16persp | ∞ | ∞ | 1035 | 2183 | 0.000 | 0.000 | 0.526 |
| netmod_dol2 | 0.047 | 0.000 | 112 | 250 | -1.000 | -1.000 | -1.000 |
| netmod_kar1 | 0.000 | 0.000 | 425 | 285 | 1.000 | 0.000 | 0.000 |
| netmod_kar2 | 0.000 | 0.000 | 275 | 285 | 1.000 | 0.000 | 0.000 |
| nous1 | 0.000 | 0.000 | 3092 | 2816 | 1.000 | 0.000 | 0.000 |
| nous2 | 0.000 | 0.000 | 81 | 71 | 1.000 | 0.000 | 0.000 |
| nuclearvb | ∞ | ∞ | 1821 | 3817 | 0.000 | 0.000 | 0.523 |
| nuclearvc | ∞ | ∞ | 1905 | 1530 | 0.000 | 0.000 | -0.197 |
| nuclearvd | ∞ | ∞ | 3781 | 2521 | 0.000 | 0.000 | -0.333 |
| nuclearve | ∞ | ∞ | 877 | 5464 | 0.000 | 0.000 | 0.839 |
| nuclearvf | ∞ | ∞ | 256 | 3596 | 0.000 | 0.000 | 0.929 |
| nvs13 | 0.000 | 0.000 | 9 | 9 | 1.000 | 0.000 | 0.000 |
| nvs17 | 0.000 | 0.000 | 89 | 78 | 1.000 | 0.000 | 0.000 |
| nvs18 | 0.000 | 0.000 | 121 | 75 | 1.000 | 0.000 | 0.000 |
| nvs19 | 0.000 | 0.000 | 161 | 154 | 1.000 | 0.000 | 0.000 |
| nvs23 | 0.000 | 0.000 | 465 | 523 | 1.000 | 0.000 | 0.000 |
| nvs24 | 0.000 | 0.000 | 2060 | 1944 | 1.000 | 0.000 | 0.000 |
| p_ball_10b_5p_2d_m | 0.000 | 0.000 | 353 | 326 | 1.000 | 0.000 | 0.000 |
| p_ball_10b_5p_3d_m | 0.000 | 0.000 | 1204 | 1032 | 1.000 | 0.000 | 0.000 |
| p_ball_10b_5p_4d_m | 0.000 | 0.000 | 1424 | 1765 | 1.000 | 0.000 | 0.000 |
| p_ball_10b_7p_3d_m | 0.000 | 0.000 | 6178 | 6151 | 1.000 | 0.000 | 0.000 |
| p_ball_15b_5p_2d_m | 0.000 | 0.000 | 1377 | 2068 | 1.000 | 0.000 | 0.000 |
| p_ball_20b_5p_2d_m | 0.000 | 0.000 | 1610 | 2039 | 1.000 | 0.000 | 0.000 |

| Name | Gap Ours | Gap Base | Nodes Ours | Nodes Base | Reward | Utility | Utility/Node |
|---|---|---|---|---|---|---|---|
| p_ball_20b_5p_3d_m | 0.000 | 0.000 | 10647 | 11510 | 1.000 | 0.000 | 0.000 |
| p_ball_30b_10p_2d_m | $\infty$ | $\infty$ | 3795 | 4965 | 0.000 | 0.000 | 0.236 |
| p_ball_30b_5p_2d_m | 0.000 | 0.000 | 2827 | 3275 | 1.000 | 0.000 | 0.000 |
| p_ball_30b_5p_3d_m | 0.000 | 0.000 | 10150 | 11489 | 1.000 | 0.000 | 0.000 |
| p_ball_30b_7p_2d_m | $\infty$ | $\infty$ | 8511 | 11906 | 0.000 | 0.000 | 0.285 |
| p_ball_40b_5p_3d_m | $\infty$ | $\infty$ | 9620 | 13718 | 0.000 | 0.000 | 0.299 |
| p_ball_40b_5p_4d_m | $\infty$ | $\infty$ | 8100 | 11826 | 0.000 | 0.000 | 0.315 |
| pedigree_ex485 | 0.019 | 0.019 | 315 | 962 | 0.030 | 0.030 | 0.682 |
| pedigree_ex485_2 | 0.000 | 0.000 | 121 | 344 | 1.000 | 0.000 | 0.000 |
| pedigree_sim400 | 0.061 | 0.053 | 1094 | 1533 | -0.156 | -0.135 | 0.175 |
| pedigree_sp_top4_250 | 0.053 | 0.036 | 61 | 173 | -0.482 | -0.325 | 0.477 |
| pedigree_sp_top4_300 | 0.014 | 0.015 | 294 | 670 | 0.014 | 0.014 | 0.567 |
| pedigree_sp_top4_350tr | 0.000 | 0.014 | 365 | 1096 | 1.000 | 0.999 | 1.000 |
| pedigree_sp_top5_250 | 0.050 | 0.057 | 28 | 39 | 0.125 | 0.125 | 0.372 |
| pinene200 | $\infty$ | $\infty$ | 12 | 12 | 0.000 | 0.000 | 0.000 |
| pointpack06 | 0.000 | 0.000 | 2099 | 2051 | 1.000 | 0.000 | 0.000 |
| pointpack08 | 0.015 | 0.000 | 35620 | 34315 | -1.000 | -0.999 | -0.978 |
| pointpack10 | 0.612 | 0.613 | 18366 | 22179 | 0.001 | 0.001 | 0.173 |
| pointpack12 | 0.854 | 0.839 | 15197 | 17796 | -0.018 | -0.018 | 0.131 |
| pointpack14 | 1.535 | 1.537 | 8919 | 9550 | 0.001 | 0.001 | 0.067 |
| pooling_adhya1pq | 0.000 | 0.000 | 383 | 365 | 1.000 | 0.000 | 0.000 |
| pooling_adhya1stp | 0.000 | 0.000 | 737 | 638 | 1.000 | 0.000 | 0.000 |
| pooling_adhya1tp | 0.000 | 0.000 | 611 | 806 | 1.000 | 0.000 | 0.000 |
| pooling_adhya2pq | 0.000 | 0.000 | 569 | 588 | 1.000 | 0.000 | 0.000 |
| pooling_adhya2stp | 0.000 | 0.000 | 832 | 934 | 1.000 | 0.000 | 0.000 |
| pooling_adhya2tp | 0.000 | 0.000 | 345 | 288 | 1.000 | 0.000 | 0.000 |
| pooling_adhya3pq | 0.000 | 0.000 | 377 | 289 | 1.000 | 0.000 | 0.000 |
| pooling_adhya3stp | 0.000 | 0.000 | 834 | 1078 | 1.000 | 0.000 | 0.000 |
| pooling_adhya3tp | 0.000 | 0.000 | 675 | 585 | 1.000 | 0.000 | 0.000 |
| pooling_adhya4pq | 0.000 | 0.000 | 274 | 150 | 1.000 | 0.000 | 0.000 |
| pooling_adhya4stp | 0.000 | 0.000 | 385 | 686 | 1.000 | 0.000 | 0.000 |
| pooling_adhya4tp | 0.000 | 0.000 | 317 | 387 | 1.000 | 0.000 | 0.000 |
| pooling_bental5stp | 0.000 | 0.000 | 2818 | 4434 | 1.000 | 0.000 | 0.000 |
| pooling_digabel16 | 0.000 | 0.000 | 27577 | 35207 | -1.000 | -0.715 | -0.160 |
| pooling_digabel18 | 0.013 | 0.008 | 4109 | 5110 | -0.496 | -0.331 | -0.168 |
| pooling_digabel19 | 0.001 | 0.001 | 14953 | 18095 | 0.168 | 0.166 | 0.267 |
| pooling_foulds2stp | 0.000 | 0.000 | 36 | 25 | 1.000 | 0.000 | 0.000 |
| pooling_foulds3stp | 0.000 | 0.000 | 1084 | 416 | 1.000 | 0.000 | 0.000 |
| pooling_foulds4stp | 0.000 | 0.000 | 717 | 339 | 1.000 | 0.000 | 0.000 |
| pooling_foulds5stp | 0.019 | 0.000 | 1808 | 2741 | -1.000 | -0.999 | -0.999 |
| pooling_haverly2stp | 0.000 | 0.000 | 10 | 12 | 1.000 | 0.000 | 0.000 |
| pooling_rt2pq | 0.000 | 0.000 | 237 | 431 | 1.000 | 0.000 | 0.000 |
| pooling_rt2stp | 0.000 | 0.000 | 109 | 195 | 1.000 | 0.000 | 0.000 |
| pooling_rt2tp | 0.000 | 0.000 | 53 | 57 | 1.000 | 0.000 | 0.000 |
| pooling_sppa0pq | 0.038 | 0.031 | 2424 | 3666 | -0.230 | -0.187 | 0.187 |
| pooling_sppa0stp | 2.829 | 2.865 | 2577 | 3068 | 0.012 | 0.012 | 0.170 |
| pooling_sppa0tp | 0.179 | 0.183 | 2804 | 3623 | 0.021 | 0.021 | 0.242 |
| pooling_sppa5pq | 0.037 | 0.018 | 709 | 781 | -0.995 | -0.499 | -0.448 |
| pooling_sppa5stp | 3.959 | 3.959 | 220 | 278 | 0.000 | 0.000 | 0.209 |
| pooling_sppa5tp | 1.579 | 1.579 | 299 | 448 | 0.000 | 0.000 | 0.333 |
| pooling_sppa9pq | 0.007 | 0.007 | 222 | 295 | 0.000 | 0.000 | 0.247 |
| pooling_sppb0pq | 0.098 | 0.098 | 223 | 301 | 0.000 | -0.000 | 0.259 |
| popdynm100 | $\infty$ | $\infty$ | 7556 | 11105 | 0.000 | 0.000 | 0.320 |
| popdynm25 | $\infty$ | $\infty$ | 14627 | 19046 | 0.000 | 0.000 | 0.232 |
| popdynm50 | $\infty$ | $\infty$ | 12085 | 15252 | 0.000 | 0.000 | 0.208 |
| portfol_classical050_1 | 0.000 | 0.000 | 651 | 817 | 1.000 | 0.000 | 0.000 |
| portfol_classical200_2 | 0.141 | 0.125 | 396 | 491 | -0.134 | -0.118 | 0.086 |
| portfol_robust050_34 | 0.000 | 0.000 | 94 | 49 | 1.000 | 0.000 | 0.000 |
| portfol_robust100_09 | 0.000 | 0.000 | 489 | 361 | 1.000 | 0.000 | 0.000 |
| portfol_robust200_03 | 0.182 | 0.189 | 95 | 75 | 0.034 | 0.034 | -0.183 |
| portfol_shortfall050_68 | 0.000 | 0.000 | 467 | 375 | 1.000 | 0.000 | 0.000 |
| portfol_shortfall100_04 | 0.010 | 0.010 | 595 | 1398 | -0.055 | -0.052 | 0.551 |
| portfol_shortfall200_05 | 0.033 | 0.028 | 224 | 232 | -0.169 | -0.145 | -0.114 |
| powerflow0009r | 0.000 | 0.000 | 15230 | 13141 | -1.000 | -0.037 | -0.003 |
| powerflow0014r | 0.001 | 0.001 | 8052 | 8041 | 0.368 | 0.366 | 0.346 |
| powerflow0030r | 0.023 | 0.034 | 369 | 403 | 0.328 | 0.328 | 0.384 |
| powerflow0039r | 0.017 | 0.016 | 212 | 224 | -0.058 | -0.054 | -0.001 |
| product | 0.028 | 0.034 | 236 | 650 | 0.197 | 0.197 | 0.708 |
| qap | 198.418 | $\infty$ | 709 | 3352 | 1.000 | 1.000 | 1.000 |
| qapw | 351.271 | $\infty$ | 874 | 2437 | 1.000 | 1.000 | 1.000 |
| qp3 | $\infty$ | $\infty$ | 29875 | 32155 | 0.000 | 0.000 | 0.071 |
| qspp_0_10_0_1_10_1 | 0.849 | 1.238 | 3860 | 3982 | 0.314 | 0.314 | 0.335 |
| qspp_0_11_0_1_10_1 | 1.071 | 1.886 | 1314 | 3036 | 0.432 | 0.432 | 0.754 |
| qspp_0_12_0_1_10_1 | 1.674 | 2.102 | 794 | 1847 | 0.204 | 0.203 | 0.658 |
| qspp_0_13_0_1_10_1 | 1.893 | 4.660 | 935 | 1380 | 0.594 | 0.594 | 0.725 |
| qspp_0_14_0_1_10_1 | 3.038 | 3.200 | 299 | 1081 | 0.050 | 0.050 | 0.737 |
| qspp_0_15_0_1_10_1 | 4.356 | 4.293 | 229 | 544 | -0.015 | -0.015 | 0.573 |

| Name | Gap Ours | Gap Base | Nodes Ours | Nodes Base | Reward | Utility | Utility/Node |
|---|---|---|---|---|---|---|---|
| ringpack_10_1 | 0.082 | 1.000 | 5346 | 6348 | 0.918 | 0.918 | 0.931 |
| ringpack_10_2 | 0.082 | 0.811 | 5402 | 6366 | 0.899 | 0.899 | 0.914 |
| ringpack_20_1 | 1.551 | 3.527 | 525 | 492 | 0.560 | 0.560 | 0.531 |
| ringpack_20_2 | 9.000 | 9.000 | 239 | 183 | 0.000 | 0.000 | -0.234 |
| ringpack_20_3 | 6.251 | 6.251 | 272 | 243 | 0.000 | 0.000 | -0.107 |
| ringpack_30_2 | 14.000 | 14.000 | 36 | 49 | 0.000 | 0.000 | 0.265 |
| sep1 | 0.000 | 0.000 | 39 | 29 | 1.000 | 0.000 | 0.000 |
| slay04h | 0.000 | 0.000 | 8 | 8 | 1.000 | 0.000 | 0.000 |
| slay04m | 0.000 | 0.000 | 7 | 7 | 1.000 | 0.000 | 0.000 |
| slay05h | 0.000 | 0.000 | 64 | 119 | 1.000 | 0.000 | 0.000 |
| slay06h | 0.000 | 0.000 | 120 | 208 | 1.000 | 0.000 | 0.000 |
| slay06m | 0.000 | 0.000 | 8 | 8 | 1.000 | 0.000 | 0.000 |
| slay07h | 0.000 | 0.000 | 420 | 952 | 1.000 | 0.000 | 0.000 |
| slay07m | 0.000 | 0.000 | 218 | 501 | 1.000 | 0.000 | 0.000 |
| slay08h | 0.000 | 0.000 | 513 | 1181 | 1.000 | 0.000 | 0.000 |
| slay08m | 0.000 | 0.000 | 193 | 554 | 1.000 | 0.000 | 0.000 |
| slay09h | 0.104 | 0.135 | 612 | 488 | 0.229 | 0.229 | 0.033 |
| slay09m | 0.000 | 0.000 | 324 | 212 | 1.000 | 0.000 | 0.000 |
| slay10h | 0.103 | 0.407 | 703 | 451 | 0.746 | 0.745 | 0.603 |
| slay10m | 0.000 | 0.000 | 3933 | 4138 | 1.000 | 0.000 | 0.000 |
| smallinvDAXr1b010-011 | 0.000 | 0.000 | 324 | 264 | 1.000 | 0.000 | 0.000 |
| smallinvDAXr1b020-022 | 0.000 | 0.000 | 657 | 906 | 1.000 | 0.000 | 0.000 |
| smallinvDAXr1b050-055 | 0.000 | 0.000 | 6083 | 4430 | 1.000 | 0.000 | 0.000 |
| smallinvDAXr1b100-110 | 0.000 | 0.000 | 15366 | 34917 | 1.000 | 0.000 | 0.000 |
| smallinvDAXr1b150-165 | 0.000 | 0.001 | 26952 | 40900 | 1.000 | 0.986 | 0.730 |
| smallinvDAXr1b200-220 | 0.000 | 0.001 | 38238 | 46021 | 0.348 | 0.342 | 0.269 |
| smallinvDAXr2b010-011 | 0.000 | 0.000 | 254 | 358 | 1.000 | 0.000 | 0.000 |
| smallinvDAXr2b020-022 | 0.000 | 0.000 | 1204 | 2016 | 1.000 | 0.000 | 0.000 |
| smallinvDAXr2b050-055 | 0.000 | 0.000 | 7868 | 6682 | 1.000 | 0.000 | 0.000 |
| smallinvDAXr2b100-110 | 0.000 | 0.000 | 12971 | 14333 | 1.000 | 0.000 | 0.000 |
| smallinvDAXr2b150-165 | 0.000 | 0.000 | 39670 | 68543 | 1.000 | 0.966 | 0.421 |
| smallinvDAXr2b200-220 | 0.000 | 0.000 | 712 | 651 | 1.000 | 0.000 | 0.000 |
| smallinvDAXr3b010-011 | 0.000 | 0.000 | 260 | 358 | 1.000 | 0.000 | 0.000 |
| smallinvDAXr3b020-022 | 0.000 | 0.000 | 1676 | 906 | 1.000 | 0.000 | 0.000 |
| smallinvDAXr3b050-055 | 0.000 | 0.000 | 5716 | 5024 | 1.000 | 0.000 | 0.000 |
| smallinvDAXr3b100-110 | 0.000 | 0.000 | 39948 | 13726 | 1.000 | 0.000 | 0.000 |
| smallinvDAXr3b150-165 | 0.000 | 0.000 | 34109 | 22132 | 1.000 | 0.000 | 0.000 |
| smallinvDAXr3b200-220 | 0.000 | 0.000 | 1078 | 433 | 1.000 | 0.000 | 0.000 |
| smallinvDAXr4b010-011 | 0.000 | 0.000 | 272 | 292 | 1.000 | 0.000 | 0.000 |
| smallinvDAXr4b020-022 | 0.000 | 0.000 | 1078 | 990 | 1.000 | 0.000 | 0.000 |
| smallinvDAXr4b050-055 | 0.000 | 0.000 | 3098 | 2666 | 1.000 | 0.000 | 0.000 |
| smallinvDAXr4b100-110 | 0.000 | 0.000 | 17899 | 26316 | 1.000 | 0.000 | 0.000 |
| smallinvDAXr4b150-165 | 0.000 | 0.000 | 32042 | 56419 | 1.000 | 0.000 | 0.000 |
| smallinvDAXr4b200-220 | 0.000 | 0.000 | 935 | 612 | 1.000 | 0.000 | 0.000 |
| smallinvDAXr5b010-011 | 0.000 | 0.000 | 242 | 381 | 1.000 | 0.000 | 0.000 |
| smallinvDAXr5b020-022 | 0.000 | 0.000 | 1798 | 884 | 1.000 | 0.000 | 0.000 |
| smallinvDAXr5b050-055 | 0.000 | 0.000 | 4276 | 3312 | 1.000 | 0.000 | 0.000 |
| smallinvDAXr5b100-110 | 0.000 | 0.000 | 37028 | 72501 | 1.000 | 0.980 | 0.570 |
| smallinvDAXr5b150-165 | 0.000 | 0.000 | 40757 | 78031 | 1.000 | 0.966 | 0.414 |
| smallinvDAXr5b200-220 | 0.000 | 0.000 | 783 | 585 | 1.000 | 0.000 | 0.000 |
| sonet22v5 | 3.752 | 2.911 | 105 | 356 | -0.289 | -0.224 | 0.620 |
| sonet23v4 | 1.407 | 1.366 | 79 | 181 | -0.030 | -0.029 | 0.550 |
| sonet24v5 | 4.070 | 3.914 | 21 | 212 | -0.040 | -0.038 | 0.897 |
| sonet25v6 | 5.161 | 4.812 | 10 | 45 | -0.072 | -0.068 | 0.762 |
| sonetgr17 | 2.252 | 2.602 | 400 | 1247 | 0.134 | 0.134 | 0.722 |
| space25 | ∞ | ∞ | 154 | 143 | 0.000 | 0.000 | -0.071 |
| spectra2 | 0.000 | 0.000 | 8 | 8 | 1.000 | 0.000 | 0.000 |
| squfl010-025 | 0.000 | 0.000 | 71985 | 75945 | 0.692 | 0.000 | 0.000 |
| squfl010-040 | 0.000 | 0.000 | 18478 | 20101 | 0.529 | 0.000 | 0.000 |
| squfl010-080 | 0.000 | 0.000 | 4509 | 8339 | 0.568 | -0.000 | 0.000 |
| squfl010-080persp | 0.000 | 0.000 | 6 | 6 | 1.000 | 0.000 | 0.000 |
| squfl015-060 | 0.000 | 0.000 | 7372 | 10223 | 0.608 | -0.000 | 0.000 |
| squfl015-060persp | 0.000 | 0.000 | 6 | 6 | 1.000 | 0.000 | 0.000 |
| squfl015-080 | 0.000 | 0.001 | 3475 | 6667 | 1.000 | 0.993 | 0.976 |
| squfl020-040 | 0.000 | 0.000 | 8358 | 10679 | 0.570 | 0.000 | 0.000 |
| squfl020-050 | 0.000 | 0.000 | 4094 | 8025 | 0.365 | -0.000 | 0.000 |
| squfl020-150 | 0.014 | 0.014 | 9 | 7 | 0.000 | 0.000 | -0.222 |
| squfl020-150persp | 0.000 | 0.000 | 16 | 16 | 1.000 | 0.000 | 0.000 |
| squfl025-025 | 0.000 | 0.000 | 15093 | 11992 | 0.997 | -0.000 | -0.000 |
| squfl025-025persp | 0.000 | 0.000 | 12 | 12 | 1.000 | 0.000 | 0.000 |
| squfl025-030 | 0.000 | 0.000 | 5523 | 14798 | 1.000 | 0.000 | 0.000 |
| squfl025-030persp | 0.000 | 0.000 | 6 | 6 | 1.000 | 0.000 | 0.000 |
| squfl025-040 | 0.000 | 0.000 | 6438 | 7860 | 0.519 | 0.000 | 0.000 |
| squfl025-040persp | 0.000 | 0.000 | 12 | 12 | 1.000 | 0.000 | 0.000 |
| squfl030-100 | 0.000 | 0.000 | 1291 | 1402 | 0.289 | 0.000 | 0.000 |
| squfl040-080 | 0.000 | 0.001 | 1034 | 1477 | 1.000 | 0.983 | 0.983 |
| squfl040-080persp | 0.000 | 0.000 | 8 | 8 | 1.000 | 0.000 | 0.000 |

| Name | Gap Ours | Gap Base | Nodes Ours | Nodes Base | Reward | Utility | Utility/Node |
|---|---|---|---|---|---|---|---|
| sssd08-04persp | 0.000 | 0.000 | 20080 | 17359 | 1.000 | 0.000 | 0.000 |
| sssd12-05persp | 0.131 | 0.133 | 63030 | 73358 | 0.016 | 0.016 | 0.154 |
| sssd15-04persp | 0.188 | 0.181 | 76121 | 77773 | -0.041 | -0.039 | -0.018 |
| sssd15-06persp | 0.285 | 0.260 | 43387 | 47623 | -0.095 | -0.087 | 0.002 |
| sssd15-08persp | 0.235 | 0.234 | 30374 | 41340 | -0.005 | -0.005 | 0.261 |
| sssd16-07persp | 0.232 | 0.214 | 41858 | 46101 | -0.086 | -0.079 | 0.014 |
| sssd18-06persp | 0.200 | 0.188 | 40346 | 48551 | -0.063 | -0.059 | 0.117 |
| sssd18-08persp | 0.383 | 0.372 | 31676 | 41841 | -0.028 | -0.027 | 0.221 |
| sssd20-04persp | 0.202 | 0.202 | 63012 | 69887 | 0.002 | 0.002 | 0.100 |
| sssd20-08persp | 0.202 | 0.195 | 27951 | 34670 | -0.036 | -0.035 | 0.164 |
| sssd22-08persp | 0.228 | 0.212 | 31593 | 33795 | -0.077 | -0.071 | -0.007 |
| sssd25-08persp | 0.178 | 0.172 | 27400 | 33837 | -0.038 | -0.037 | 0.159 |
| st_bsj2 | 0.000 | 0.000 | 17 | 15 | 1.000 | 0.000 | 0.000 |
| st_e05 | 0.000 | 0.000 | 59 | 75 | 1.000 | 0.000 | 0.000 |
| st_e24 | 0.000 | 0.000 | 7 | 7 | 1.000 | 0.000 | 0.000 |
| st_e25 | 0.000 | 0.000 | 15 | 15 | 1.000 | 0.000 | 0.000 |
| st_e30 | 0.000 | 0.000 | 47 | 61 | 1.000 | 0.000 | 0.000 |
| st_e31 | 0.000 | 0.000 | 593 | 490 | 1.000 | 0.000 | 0.000 |
| st_fp7a | 0.000 | 0.000 | 297 | 345 | 1.000 | 0.000 | 0.000 |
| st_fp7b | 0.000 | 0.000 | 349 | 341 | 1.000 | 0.000 | 0.000 |
| st_fp7c | 0.000 | 0.000 | 253 | 449 | 1.000 | 0.000 | 0.000 |
| st_fp7d | 0.000 | 0.000 | 277 | 355 | 1.000 | 0.000 | 0.000 |
| st_fp7e | 0.000 | 0.000 | 1605 | 1831 | 1.000 | 0.000 | 0.000 |
| st_fp8 | 0.000 | 0.000 | 69 | 63 | 1.000 | 0.000 | 0.000 |
| st_glmp_ss1 | 0.000 | 0.000 | 23 | 25 | 1.000 | 0.000 | 0.000 |
| st_ht | 0.000 | 0.000 | 13 | 11 | 1.000 | 0.000 | 0.000 |
| st_iqpbk1 | 0.000 | 0.000 | 37 | 37 | 1.000 | 0.000 | 0.000 |
| st_iqpbk2 | 0.000 | 0.000 | 39 | 37 | 1.000 | 0.000 | 0.000 |
| st_jcbpaf2 | 0.000 | 0.000 | 9 | 13 | 1.000 | 0.000 | 0.000 |
| st_m1 | 0.000 | 0.000 | 783 | 383 | 1.000 | 0.000 | 0.000 |
| st_m2 | 0.000 | 0.000 | 637 | 619 | 1.000 | 0.000 | 0.000 |
| st_pan1 | 0.000 | 0.000 | 11 | 11 | 1.000 | 0.000 | 0.000 |
| st_ph11 | 0.000 | 0.000 | 11 | 11 | 1.000 | 0.000 | 0.000 |
| st_ph12 | 0.000 | 0.000 | 13 | 13 | 1.000 | 0.000 | 0.000 |
| st_ph13 | 0.000 | 0.000 | 9 | 9 | 1.000 | 0.000 | 0.000 |
| st_qpc-m1 | 0.000 | 0.000 | 15 | 17 | 1.000 | 0.000 | 0.000 |
| st_qpc-m3a | 0.000 | 0.000 | 1269 | 1291 | 1.000 | 0.000 | 0.000 |
| st_qpk1 | 0.000 | 0.000 | 7 | 7 | 1.000 | 0.000 | 0.000 |
| st_qpk2 | 0.000 | 0.000 | 27 | 27 | 1.000 | 0.000 | 0.000 |
| st_qpk3 | 0.000 | 0.000 | 137 | 133 | 1.000 | 0.000 | 0.000 |
| st_rv1 | 0.000 | 0.000 | 107 | 81 | 1.000 | 0.000 | 0.000 |
| st_rv2 | 0.000 | 0.000 | 133 | 119 | 1.000 | 0.000 | 0.000 |
| st_rv3 | 0.000 | 0.000 | 511 | 629 | 1.000 | 0.000 | 0.000 |
| st_rv7 | 0.000 | 0.000 | 1143 | 1153 | 1.000 | 0.000 | 0.000 |
| st_rv8 | 0.000 | 0.000 | 1047 | 1269 | 1.000 | 0.000 | 0.000 |
| st_rv9 | 0.000 | 0.000 | 3349 | 1875 | 1.000 | 0.000 | 0.000 |
| st_testgr1 | 0.000 | 0.000 | 38 | 21 | 1.000 | 0.000 | 0.000 |
| st_z | 0.000 | 0.000 | 9 | 9 | 1.000 | 0.000 | 0.000 |
| supplychain | 0.000 | 0.000 | 119 | 95 | 1.000 | 0.000 | 0.000 |
| tln12 | 0.295 | 0.217 | 20517 | 22942 | -0.362 | -0.266 | -0.179 |
| tln4 | 0.000 | 0.000 | 13 | 25 | 1.000 | 0.000 | 0.000 |
| tln6 | 0.000 | 0.000 | 40 | 38 | 1.000 | 0.000 | 0.000 |
| tln7 | 0.075 | 0.121 | 52425 | 60523 | 0.375 | 0.375 | 0.457 |
| toroidal3g7_6666 | 0.200 | 0.117 | 51 | 213 | -0.706 | -0.414 | 0.592 |
| tricp | $\infty$ | $\infty$ | 275 | 356 | 0.000 | 0.000 | 0.228 |
| util | 0.000 | 0.000 | 48 | 38 | 1.000 | 0.000 | 0.000 |
| wastewater02m1 | 0.000 | 0.000 | 43 | 43 | 1.000 | 0.000 | 0.000 |
| wastewater02m2 | 0.000 | 0.000 | 35 | 31 | 1.000 | 0.000 | 0.000 |
| wastewater04m1 | 0.000 | 0.000 | 117 | 81 | 1.000 | 0.000 | 0.000 |
| wastewater04m2 | 0.000 | 0.000 | 25 | 25 | 1.000 | 0.000 | 0.000 |
| wastewater05m1 | 0.000 | 0.000 | 2561 | 3047 | 1.000 | 0.000 | 0.000 |
| wastewater05m2 | 0.000 | 0.000 | 4068 | 7429 | 1.000 | 0.000 | 0.000 |
| wastewater11m1 | 0.116 | 0.131 | 40219 | 43385 | 0.113 | 0.113 | 0.177 |
| wastewater11m2 | 0.385 | 0.431 | 15161 | 15304 | 0.106 | 0.106 | 0.114 |
| wastewater12m1 | 0.099 | 0.045 | 23070 | 28082 | -1.000 | -0.541 | -0.440 |
| wastewater12m2 | 0.460 | 0.654 | 7232 | 7822 | 0.296 | 0.296 | 0.349 |
| wastewater13m1 | 0.446 | 0.370 | 12150 | 16381 | -0.207 | -0.171 | 0.105 |
| wastewater13m2 | 0.538 | 0.538 | 6204 | 6129 | 0.000 | 0.000 | -0.012 |
| wastewater14m1 | 0.151 | 0.122 | 38064 | 42510 | -0.236 | -0.191 | -0.096 |
| wastewater14m2 | 0.191 | 0.209 | 11743 | 13355 | 0.084 | 0.084 | 0.194 |
| wastewater15m1 | 0.000 | 0.000 | 7735 | 8130 | 1.000 | 0.000 | 0.000 |
| wastewater15m2 | 0.000 | 0.000 | 54228 | 59163 | 0.982 | -0.000 | -0.000 |
| watercontamination0303 | 0.000 | 0.000 | 9 | 9 | 1.000 | 0.000 | 0.000 |
| watercontamination0303r | $\infty$ | $\infty$ | 22 | 37 | 0.000 | 0.000 | 0.405 |
| waterund01 | 0.000 | 0.000 | 49001 | 57176 | -0.022 | -0.021 | 0.056 |
| waterund08 | 0.000 | 0.000 | 38355 | 41489 | 0.335 | 0.083 | 0.003 |
| waterund11 | 0.001 | 0.001 | 35021 | 40695 | -0.736 | -0.420 | -0.241 |

| Name | Gap Ours | Gap Base | Nodes Ours | Nodes Base | Reward | Utility | Utility/Node |
|---|---|---|---|---|---|---|---|
| waterund14 | 0.009 | 0.009 | 9789 | 10684 | -0.012 | -0.012 | 0.072 |
| waterund17 | 0.001 | 0.001 | 35708 | 36527 | 0.549 | 0.545 | 0.436 |
| waterund18 | 0.001 | 0.001 | 34080 | 36286 | 0.049 | 0.048 | 0.085 |
| waterund22 | 0.016 | 0.017 | 10195 | 10702 | 0.016 | 0.016 | 0.062 |
| waterund25 | 0.080 | 0.094 | 11100 | 10382 | 0.154 | 0.153 | 0.095 |
| waterund27 | 0.089 | 0.089 | 2253 | 2835 | 0.001 | 0.001 | 0.206 |
| waterund28 | 0.080 | 0.080 | 18 | 17 | 0.000 | 0.000 | -0.056 |
| waterund36 | 0.100 | 0.082 | 1841 | 2443 | -0.217 | -0.178 | 0.083 |
| Mean | — | — | 6315 | 7463 | **0.487** | 0.000 | **0.114** |

### D.3.1 KOCHETOV-UFLP

To demonstrate the generalizability of the learned heuristics, we test our method on the Uncapacitated Facility Location Problem (see Appendix B) *without further finetuning*, i.e., we only train on TSP instances and never show the algorithm any other linear or nonlinear problem. For testing, we generate 1000 instances using the well-known problem generator by Kochetov & Ivanenko (2005), which was designed to have large optimality gaps, making these problems particularly challenging.

Our method performs very similar to the highly optimized baseline, despite never having seen the UFL problem, see Table 6. We argue that this is specifically because our method relies on tree-wide behaviour, rather than individual features to make decisions. We further hypothesize that the reason for the advantage over the baseline being so small is due to the fact that UFLP consists of "adversarial examples" to the branch-and-bound method where cuts have reduced effectiveness. This means clever node-selection strategies have limited impact on overall performance.

An interesting aspect is that our method processes more nodes than the baseline, which also leads to the loss in node-efficiency. This implies that our method selects significantly easier nodes, as ordinarily our solver is slower just due to the additional overhead. Considering that this benchmark was specifically designed to produce high optimality gaps, it makes sense that our solver favours node quantity over quality, which is an interesting emergent behaviour of our solver.

## E ARCHITECTURE

Our network consists of two subsystems: First, we have the feature embedder that transforms the raw features into embeddings, without considering other nodes this network consists of one linear layer $|d_{features}| \rightarrow |d_{model}|$ with LeakyReLU (Xu et al., 2015) activation followed by two $|d_{model}| \rightarrow |d_{model}|$ linear layers (activated by LeakyReLU) with skip connections. We finally normalize the outputs using a Layernorm (Ba et al., 2016) *without* trainable parameters (i. e., just shifting and scaling the feature dimension to a normal distribution).

Second, we consider the GNN model, whose objective is the aggregation across nodes according to the tree topology. This consists of a single LeakyReLU activated layer with skip-connections. We use ReZero (Bachlechner et al., 2020) initialization to improve the convergence properties of the network. Both the weight and value heads are simple linear projections from the embedding space. Following the guidance in (Andrychowicz et al., 2020), we make sure the value and weight networks are independent by detaching the value head's gradient from the embedding network.

## F NECESSITY OF GNN

In addition to the tests done above, we also investigated running the model without a GNN: We found that when removing the GNN, the model tended to become very noisy and produce unreproducible experiments. Considering only the cases where the GNN-free model did well, we still found the model needed roughly 30% more nodes than the SCIP or our model with a GNN. More importantly, we notice the GNN-free model diverges during training: starting with a reward of roughly zero, the model diverges down to a reward of $\approx -0.2$, which amounts to a score roughly 20% worse than SCIP. We therefore conclude that, at least for our architecture, the GNN is necessary for both robustness and performance.

## G   RESULTS ON TRAINING SET

Table 6: Performance on the training set.

| Benchmark | Reward | Utility | Utility/Node | Win-rate | geo-mean Ours | geo-mean SCIP |
|---|---|---|---|---|---|---|
| Training set | 0.102 | 0.116 | 0.383 | 0.76 | 0.506 | 0.582 |

