# OpenReview forum: "Reinforcement Learning for Node Selection in Branch-and-Bound"
_ICLR.cc/2024/Conference — Submitted to ICLR 2024_

### Official Review · Reviewer_Lp54 · 2023-10-30

**Soundness:** 2 fair
**Presentation:** 2 fair
**Contribution:** 3 good
**Rating:** 3
**Confidence:** 4

**Summary:**

Authors propose a reinforcement learning algorithm for node selection problem in Branch-and-Bound (B&B) for Mixed Integer Programming. While prior work mostly focuses on ranking a pair of nodes, authors propose to use GNN to leverage information across the B&B tree. The policy induces a distribution across all open nodes in B&B tree. Authors propose Policy network and Value network architecture based on GNN. The proposed architecture is trained on TSP problems, and evaluated on both TSP and MIPLIB benchmark. The proposed method outperforms SCIP's default node selector across benchmarks considered.

**Strengths:**

Significance: Node Selection is instrumental for successful B&B. Prior research has been focused on Imitation Learning. This is limiting because when "expert" policy doesn't work well on problem at hand, ML-based method similarly struggles. Therefore, authors' Reinforcement Learning-based method has the potential of allowing ML-based node selection methods to be applied to a broader range of mixed integer programming problems with bigger, more practical improvements. Hence, I consider the potential significance to be high.

Originality: Given a related problem of Variable Selection has gone through a similar transition from Imitation Learning to Reinforcement Learning, the proposal of reinforcement learning method is not entirely unforeseen. However, authors make original contribution by proposing how to represent states of Markov Decision Process with GNNs.

**Weaknesses:**

Quality: Experimental setup of the paper could be improved more directly test the paper's key hypothesis: that 1) Reinforcement Learning provides an advantage over Imitation Learning, 2) considering the entire tree state is better than just considering isolated nodes. These are points which distinguish authors' work from prior work. Unfortunately, authors compare against only SCIP's default node selector, and previously proposed algorithms are not considered.

Also, authors use metrics and benchmark datasets not used in previous papers in this area of research. This makes difficult to interpret experimental results within the context of current research. In fact, many of the issues with metrics authors run into could be addressed with Primal/Dual/Gap Integral metrics https://www.ecole.ai/2021/ml4co-competition/ (see Metrics page), as these metrics would still be sensible when one algorithm can reduce the gap to be zero; since authors' metrics are not very well-defined when zero gap can be (nearly) reached, authors had to employ nontrivial preprocessing of data.

These two are major concerns. The contribution of the paper is mostly the proposal of an empirical method that improves upon prior work, and therefore it is important for experiments to be designed to measure the advantage of the proposed method upon prior art.

Clarity: The main ideas of the paper is clearly described and easy to follow. Some technical statements did not provide sufficient reasoning to justify, however. For example, in Section 4.3, it's argued: Assuming $P \neq NP$, it is unreasonable for the proposed algorithm to tackle provably hard instances. In practice, MIP solvers are often applied to problems which don't allow even good approximation guarantee, and therefore I wasn't sure why $P \neq NP$ would imply these problems to be not tractable.

**Questions:**

Is the reward (equation 5) only received at the end of the "episode" (end of the MIP solve)?

---

> ### Author Response · Authors · 2023-11-11
>
> First and foremost, we thank the reviewer for his time. We thank the reviewer for noting our improvements on one of the core aspects of the branch-and-bound algorithm.
>
> > 1) Reinforcement Learning provides an advantage over Imitation Learning,
>
> It is not really possible for our methods to use an off-the-shelf imitation learning approach due to one of our (implicit) core assumptions: The reinforcement learner is going to “discover” novel problems during search which were not originally specified, i.e. while we only generate TSP instances, due to the branching and cutting inside the algorithm, the model will still observe sub-trees which have significantly different structure. This is why we can get away with training only on 200 synthetic instances, while prior art often uses 100k across different problems. However, this also means that imitation learning algorithms are going to get stuck (if they work well) as they are not going to explore as much. Further, using off-the-shelf imitation learning models is not possible due to how the model represents the environment:
> For instance, which future actions a model has fully depends on which past actions were performed, i.e. you have #leaves “actions”, and which actions you have available depends on the prior actions. This means behavioral cloning by e.g. minimizing the KL-divergence between agent and expert distributions is simply not possible, because after two or three steps the distributions between them will be almost fully disjoint:
> Even the domain of the expert’s probability density function (i.e. set of leaves) might be different from what the agent explored (meaning that e.g. the KL-divergence would always be infinite).
> This does not mean imitation learning is fully impossible, but it does mean that imitation learning would need specific algorithmic adaptations to work in this setting.
> In that sense the advantage of Reinforcement Learning over Imitation Learning is that this policy is trainable _at all_ using standard techniques.
>
> We have clarified this in our methods section.
> Were we able to clarify your concerns?
>
>
> > 2) considering the entire tree state is better than just considering isolated nodes. These are points which distinguish authors' work from prior work. Unfortunately, authors compare against only SCIP's default node selector, and previously proposed algorithms are not considered.
>
> Just considering isolated nodes it is not possible to design a node selection policy as node selection specifically relies on the relative difference between the quality of two nodes. This is why Labassi et al had to phrase their "node local" representation as a comparison between two nodes as individual nodes are fundamentally not capable of selecting their own priority in isolation.
> The closest thing we can try is setting our message-passing iterations to zero, which means that only the pathwise sums are used to build the tree representations. While these are still global representations, this would allow one to test whether the quality of the non-local embeddings has significant effects on the overall policy's quality.
> Thank you for discussing this point, we will run these experiments and update the paper as soon as the results are ready (soon).
>
> You further note that a comparison against Labassi et al would have been useful:
> This is true, however benchmarking other learned node selectors proved challenging: The major piece of prior art is Labassi et al’s “Learning to Compare Nodes in Branch and Bound with Graph Neural Networks”. Unfortunately this method is hard-coded on SCIP version 7, while we rely on SCIP version 8. Considering that “The SCIP Optimization Suite 8.0” (https://or.rwth-aachen.de/files/research/repORt/scipopt-80.pdf) estimates the difference between SCIP 7 and SCIP 8 at up-to 50%, meaning that labassi’s method loses simply due to being implemented in a 2 year old MILP framework (as a side note: Labassi also only works for Linear Problems, so would not even apply to our nonlinear benchmarks).
> Considering that SCIP has been the most dominant open-source MILP solver for over a decade (see e.g. https://plato.asu.edu/bench.html), we think that benchmarking against it is a fair representation of the overall performance (i.e. we consider the SOTA MILP solver, rather than the SOTA under the Deep Learning heuristics specifically).

---

> > ### Author Response · Authors · 2023-11-11
> >
> > > This makes difficult to interpret experimental results within the context of current research. In fact, many of the issues with metrics authors run into could be addressed with Primal/Dual/Gap Integral metrics https://www.ecole.ai/2021/ml4co-competition/ (see Metrics page), as these metrics would still be sensible when one algorithm can reduce the gap to be zero; since authors' metrics are not very well-defined when zero gap can be (nearly) reached, authors had to employ nontrivial preprocessing of data.
> >
> > Thank you for the reference towards these additional metrics: However, I’m not sure that these are applicable for us, since they assume a trivial primal solution as a reference (which is not guaranteed for us, since the benchmark sets contain e.g. satisfaction problems where finding any primally valid solution makes up the entire problem), and do not work in nonlinear problems (in the sense that the optimal primal-dual gap is nonzero at the optimum and may be arbitrarily large depending on the instance).
> > This would certainly be very useful for using as a reward though, if we selected our dataset to have this well defined at the root!
> > In general, we would also argue that reward shaping really is an orthogonal problem to the model’s design: Our objective was to build a reward that works under the considerations of our dataset selection. If one wanted to generalize this then framing the rewards as integral metrics would certainly be interesting.
> >
> > To make our work more comparable, we decided to also post the shifted geometric means of the optimality gaps as these are currently used for comparing SOTA MILP and MINLP solvers, which can be seen as evaluating the normalized Primal-Dual gap at the end of the solve.
> >
> > > Assuming P!=NP, it is unreasonable for the proposed algorithm to tackle provably hard instances. In practice, MIP solvers are often applied to problems which don't allow even good approximation guarantee, and therefore I wasn't sure why P!=NP would imply these problems to be not tractable.
> >
> > You are right, this was formulated confusingly. Allow me to rephrase this:
> > What we were referencing is that blindly generating hard instances would not lead to a useful policy. E.g. we could generate knapsack instances according to the Merkle–Hellman knapsack cryptosystem, and while these would certainly be more complex than simple random instances (which have frequently been shown to have a low expected complexity), it would also not produce a good node selection policy as the problem is too hard.
> > I.e. while random instances are too easy, blindly searching for hard instances will only yield instances which are horrible for the learner.
> > This doesn’t mean that our method could not be applied to these types of instances (in fact the UFLP instances are precisely ones designed to be hard for Branch-and-Bound), but it means these instances are suboptimal for learning.
> >
> >
> > > Is the reward (equation 5) only received at the end of the "episode" (end of the MIP solve)?
> > It is indeed only received at the end of a solve (we updated the paper accordingly).
> > This is done because we do not assume that the solver is able to find a primally valid bound at the root node (which for our training data would already be a hamilton cycle). This is also the big hurdle against testing the primal-dual gap as a reward: The instance generation process does not make the assumption that this primal-dual gap is finite at any point.
> >
> >
> > We hope we were able to address your concerns and are happy to answer further questions.

---

> ### Author Response · Authors · 2023-11-15
>
> Once again, thank you for your remarks regarding local vs global representations!
> We conducted additional experiments by training an additional agent without the GNN (i.e. zero message passing steps), to test the performance in a “minimally global” setting.
> Note that completely independent node predictors do not work, as without correlating nodes at least pairwise, no node can determine whether it is “the most optimal” simply because the node does not know which other options exist.
>
> Testing the model without message passing, gave some interesting insights.
> Firstly, the model without the GNN is considerably more noisy; so much so that benchmarking on our tests sets does not really make sense: Running the same model 3 times on TSPLIB essentially yields 3 different results.
> Considering only the cases where the GNN-free model reaches low optimality gaps, the model needed considerably more nodes than the baseline or our model with GNN to reach comparable performance (roughly 30%, but that number is prone to noise).
> Secondly, models trained without a GNN diverge on the training set: Starting off at a reward of roughly zero, it diverges to a reward of -0.2 (i.e. 20% worse than SCIP) by the end of training.
> We have added these additional details in appendix F.
>
> Thank you once again for your pointer and we would be excited to answer additional questions!

---

### Official Review · Reviewer_rRw3 · 2023-10-31

**Soundness:** 3 good
**Presentation:** 2 fair
**Contribution:** 2 fair
**Rating:** 5
**Confidence:** 3

**Summary:**

This paper tackles the brand-and-bound problem, for which the authors propose a new method of simulation using RL for getting a more global view of the tree, augmented with heuristic node selection methods: tree encoding by GNN, features are learned by message passing and node selection is done by PPO. Experiments show positive results on many benchmarks despite the training being on TSP simulations. Another good thing is that code is provided (although I haven’t tested myself).

**Strengths:**

- The global view of trees is a strong motivation given the limitation of current methods in BnB.
- I personally like the “greedy” aspect in reasoning (in introduction) that theory vs. practice has a gap, especially for many cases like the BnB and in practice, oftentimes we should favor a shorter-term choice over long-term ones if it’s good enough for many reasons. I think that is correct to the large spectrum of deep learning applications nowadays.
- Positive results on many benchmarks.
- Helpful supplemental contents.

**Weaknesses:**

- The strong motivation leads to a much larger cost in carrying out the algorithm, especially when it involves recursion.. However, it’s not clear from the paper as to why the authors only choose the upper bound as a factor of choosing. Would be interesting if they have a study –of maybe a comparison–leading to that choice.
- To solve this complex problem, the proposed method has to be broken down into many phases as shown in Section 2. That raises a question about the practicality: can the method be integrated as one to make it end-to-end. If not yet, what are the factors needed or what changes to enable that.
- Another unclear aspect is the design of the reward method, e.g. why that formula in terms of motivation and explanation, and why not replace the term (“-1”) in Equation 5 with a constant C and study different values of it?
- After the reward function, yet another unexplained technique of “shifting”, and another heuristics of clipping the reward. Is there any other better way of normalizing that or better design of the reward function to make sure that range complies while having a nice curve to the problem?
- Why PPO? Would also be nice if comparing PPO to alternatives such as maybe TRPO, SAC, …
- Yet another heuristics is to remove problems >100% or 0 gap. That begs a question on the quality of design including the reward function.
- Overall, the paper gives an impression that despite a good motivation and a complex problem, it’s a collection of heuristic choices without substantiated evidence/studies supporting them. Such heuristics I think undermine the main motivation (i.e. ones might question how much contribution of RL in yielding the results you are getting?) It would be much more convincing if the authors address this aspect now or later.

======

Additionally: some monor typos:
- Maybe should not write TSP in the abstract as abbreviation (abbr), which is inconsistent, since the abbr term RL was defined before that.
- Just in case, please use the newest ICLR 2024 template
- Appendix E citation format is not consistent with other parts
- Section3, first line of 2nd paragraph, define the abbr “IL” first.

**Questions:**

- As also stated and shown, the problems are hard to handle computationally due to numerical instability. It is however not clear what problems they run into, and how the authors handle them. Those are very important for the community in terms of insights and reproducibility.
- Table 3: The “Gap Ours” and “Gap Base” column have all normal values but the mean is NaN. Why?
- See other questions in the Weaknesses section.

---

> ### Author Response · Authors · 2023-11-11
>
> We first and foremost want to thank the reviewer for his time. Thank you for highlighting the strengths of our paper in the form of global tree representation, our thorough benchmarking and supplemental information.
>
> > The strong motivation leads to a much larger cost in carrying out the algorithm, especially when it involves recursion.. However, it’s not clear from the paper as to why the authors only choose the upper bound as a factor of choosing. Would be interesting if they have a study –of maybe a comparison–leading to that choice.
>
> Due to the fact we do not train our model during inference (solving the benchmarking instances), the actual overhead is relatively small and mostly exists due to python being slower than optimized C code.
> We will clarify this in our revision.
> We could also parameterize the selection of leaves as a comparison, but that would ultimately yield a very similar objective. Ultimately we select max(leaves) one could also rephrase this as selecting the leaf such that it is the most optimal one wrt node comparison, but ultimately this would produce the objective “choose x such that x >= y for all leaves y”, which in turn amounts to maximization.
> We mostly chose to select based on maximum probability as it aligns well with prior research into reinforcement learning (i.e. the same as action selection in other environments as well).
>
> > To solve this complex problem, the proposed method has to be broken down into many phases as shown in Section 2. That raises a question about the practicality: can the method be integrated as one to make it end-to-end. If not yet, what are the factors needed or what changes to enable that.
>
> The graphic may be a little misleading here: The model is trained end-to-end. One has to think of SCIP as the RL “environment” which, just like all other RL environments , is thought of as a black box of input actions and output states. The actual selection policy is still trained end-to-end, just like in all other RL problems.
>
> > Another unclear aspect is the design of the reward method, e.g. why that formula in terms of motivation and explanation, and why not replace the term (“-1”) in Equation 5 with a constant C and study different values of it?
>
> The movement by -1 is simply mean-shifting the reward, just like one would do in a normal regression problem. Clipping has been a common technique for reward stabilization for a number of years (see e.g. Human-level control through deep reinforcement learning https://www.nature.com/articles/nature14236), especially in environments with a high possible range of rewards (such as Atari). We will make sure to reference this.
>
> > Why PPO? Would also be nice if comparing PPO to alternatives such as maybe TRPO, SAC, …
> We chose PPO due to its relative robustness and easy hyperparameter tuning. Off-policy methods like SAC are theoretically very appealing, but end up being hard to tune properly.
> As our method relies on a novel network design in a relatively under researched field, we opted to err on the side of robustness.
>
> > Yet another heuristics is to remove problems >100% or 0 gap. That begs a question on the quality of design including the reward function.
>
> This is simply a restriction one has when working with mixed-integer solvers such as SCIP: For some instances, SCIP’s solver and simplification routines are simply so advanced that the instance becomes trivial. At that point, one could keep this instance in the training set, but that would give misleading rewards to the model since even horrible selections could solve these instances simply due to the power of SCIP itself. On the other hand, there are problems that are so hard one can never make any progress on them, even if you had oracle node-selections. To improve the signal-to-noise ratio of our reward, we decided to filter those out proactively. This is presumably also one of the reasons why we can train a highly generalizing agent with only 200 instances, compared to e.g. Labassi which needed over 100k training instances across different problems.
>
> > Additionally: some monor typos [...]
> Thank you for the heads up!
> We will fix this as soon as we update the paper with the additional results mentioned for Reviewr hSJj.

---

> > ### Author Response · Authors · 2023-11-11
> >
> > > As also stated and shown, the problems are hard to handle computationally due to numerical instability. It is however not clear what problems they run into, and how the authors handle them. Those are very important for the community in terms of insights and reproducibility.
> >
> > We’ve not had major problems with numerical instability during training, but we did have issues due to computational overhead of our method compared to the SCIP baseline:
> > When benchmarking against SCIP one has to consider that SCIP is a highly optimized piece of software that was created over a time horizon of over two decades. This means that any research prototype one writes is going to have trouble competing with this level of optimization. However, we found that at least for our method actively benchmarking on a reference instance and measuring the runtime of individual functions, then optimizing accordingly, was able to make this not have too much of an impact.
> >
> > We thank the reviewer for his excellent points in helping us clarify the paper's contents. We would be happy to discuss further in case additional questions arise.

---

> > > ### Comment · Reviewer_rRw3 · 2023-11-18
> > >
> > > Post rebuttal: after reading others' reviews and authors' responses, I appreciate the authors for the great effort in responding and making revisions. I think the revised version is a lot more clear. Yet I hope the paper will have another round of revision to highlight more an underscored theme of RL with a global tree view, rather than a mixture of heuristics to pull the main method off.
> > > For now, I am keeping my scores, thinking it's almost there.

---

### Official Review · Reviewer_KqYv · 2023-11-01

**Soundness:** 2 fair
**Presentation:** 2 fair
**Contribution:** 1 poor
**Rating:** 3
**Confidence:** 3

**Summary:**

This paper proposes a novel method for node selection in branch-and-bound using reinforcement learning. The proposed method uses a graph neural network to model the node selection as a probability distribution considering the entire tree. Based on this, reinforcement learning is applied to perform node selection.

**Strengths:**

1. The paper clearly states the issue (node selection in branch-and-bound) trying to address, and the limitation of the conventional methods on that issue.
2. The paper provides the simulation results in a variety of problem instances.

**Weaknesses:**

1. There are existing related works that use graph neural networks for node selection in the branch-and-bound algorithm. The proposed method in this paper uses graph neural networks for tree representation, but the difference from the existing works is not clearly stated.
2. The structure of RL such as states, actions, and reward function is not rigorously defined in the paper. This makes it harder to understand how the RL method works in the proposed method.
3. As the branch-and-bound algorithm proceeds, the number of nodes and the tree structure keep changing. Then, should the RL agent be trained from the scratch for every step of the branch-and-bound algorithm?
4. The discussion on the learning cost of the RL algorithm is required. How is the cost due to collecting the enough experiences for the convergence of the RL policy?
5. It seems that the RL agent should be trained for each problem instance. Is this training of RL agent for each problem instance is mandatory to use the proposed method? If additional RL agent training is always required, it may not be practical to use it.
6. In the similar context, is there a possibility of using a pretrained RL agent that can be applied to a variety of problems? It would be helpful to demonstrate the generalization capabilities of the RL agent for more insight on the proposed method.
7. Comparison with related works using graph neural networks, in particular, Labassi et al. (2022) which is one of the state-of-the-art methods, seems to be necessary in experiments. The authors stated that it was unable to conduct experiments due to the version compatibility issue, but a comparison between the similar state-of-the-art methods is essential.

**Questions:**

Please refer to the weaknesses.

---

> ### Author Response · Authors · 2023-11-11
>
> First and foremost, we thank the reviewer for his time and effort and thank you for recognizing our main contributions.
> > There are existing related works that use graph neural networks for node selection in the branch-and-bound algorithm. The proposed method in this paper uses graph neural networks for tree representation, but the difference from the existing works is not clearly stated.
>
> We are not aware of any method that utilizes Graph Neural Networks to represent the branch and bound tree.
> Do you have references for relevant works?
> We are only aware of Graph representations of individual nodes in the tree, i.e. every node is one constraint graph between the node’s variables and constraints. One contemporary work published during this review cycle also uses an implicit tree representation, where instead of a timeseries the authors consider a “Tree series” where the MDP is modelled as a tree rather than a linear timeseries. They also consider the variable selection problem, rather than the node selection problem.
>
> Both of these approaches are completely orthogonal to ours: Our approach allows for TreeMDPs and nodes being represented as Graphs just the same.
>
> However, in some sense our approach can be seen as a transpose of the constraint-graph representation: Constraint graphs consider one node and anchor their graph to the instance, while our approach considers the entire graph and anchors its graph to the branch and bound structure.
> Our approach does not need this implicit correspondence between a nodes constraint matrix and graph, which allows us to tackle nonlinear problems in which no constraint matrix exists.
>
> Does the reviewer think it would be useful for us to make this connection as part of the paper?
>
> > The structure of RL such as states, actions, and reward function is not rigorously defined in the paper. This makes it harder to understand how the RL method works in the proposed method.
>
> We amended our paper to define this more thoroughly by giving the tuple of (State, Action, Reward) explicitly.
>
> > As the branch-and-bound algorithm proceeds, the number of nodes and the tree structure keep changing. Then, should the RL agent be trained from the scratch for every step of the branch-and-bound algorithm?
>
> This is not a problem due to how we design our model: Each state is an entire Tree that is parameterized as a graph neural network. After each state transition we still have a graph, which can be processed just the same using the same network and the same weights. The trick behind our algorithm is precisely that we can obtain a global view on the entire problem _without_ needing any modifications to the network (similar to how an LSTM works regardless of sequence length, our GNN works regardless of tree size and topology).
>
> Bundling the next remarks into one:
> > The discussion on the learning cost of the RL algorithm is required. How is the cost due to collecting the enough experiences for the convergence of the RL policy? It seems that the RL agent should be trained for each problem instance. Is this training of RL agent for each problem instance is mandatory to use the proposed method? [...]
>
> All experiments in this paper were performed using the same agent trained only on synthetic samples. This means that during benchmarking no training takes place at all: It just infers the correct next node by using the existing fixed weights. Overall our method was only ever trained on a small dataset of 200 instances of the traveling salesmen problem for what amounts to 16 epochs (generation details are in section 4.3).
> Specifically, the benchmarks are all done using a _single pretrained_ agent: The agent was trained once on the synthetic instances, the weights are frozen and then the agent is run on the different benchmark datasets.
> These datasets are substantially more complex, larger, and of completely different problems than what can be found in the training dataset.
> In practice, this would amount to e.g. SCIP being delivered with a pretrained policy which will be automatically used without additional training on novel problems.
>
> Other reviewers also noted this confusion, hence we updated the paper to make this more clear (see abstract)

---

> > ### Author Response · Authors · 2023-11-11
> >
> > Regarding Labassi et al: We ran into issues due to the fact that their method only works using linear problems, while we test both linear and nonlinear instances.We simply cannot run Labassi’s method on our benchmark sets since we consider a more general problem. Further, their tests were hard-coded on a two and a half year old version of SCIP, which makes it even more impossible to test, and means that the results presumably are no longer SOTA due to the continuous large improvements in mixed integer programming: The whitepaper for SCIP “The SCIP Optimization Suite 8.0” (https://or.rwth-aachen.de/files/research/repORt/scipopt-80.pdf) showcases an up-to 50% performance improvement of SCIP 8 compared to the old SCIP 7. Therefore, benchmarking against Labassi is simply not sensible as the difference between SCIP 8 and SCIP 7 alone is larger than Labassi’s improvement over stock SCIP 7.
> > Due to significant changes between SCIP 8 and SCIP 7, it is not feasible to run our method on the older SCIP version.
> >
> > SCIP itself has shown to be the most dominant open-source Mixed Integer solver (see, e.g. https://plato.asu.edu/bench.html for benchmarks) for close to a decade, meaning that measuring performance against it should give a good comparison against the state-of-the-art.
> >
> > We hope we were able to address your concerns and hope to further discuss with you.

---

> > > ### Comment · Reviewer_KqYv · 2023-11-20
> > >
> > > After reading the responses, some of the concerns have been addressed and the revised paper is improved. However, two primary concerns are still not addressed.
> > >
> > > 1) I still don't feel that the RL structure is rigorously defined. The revision is careless and the MDP is still not well defined mathematically. The additional descriptions are not clear and make it even more confusing.
> > > 2) Comparison with other SOTA algorithms is essential, and the authors' response cannot be an excuse for not doing experiments. SCIP is just one of the solvers, and the existing algorithms typically present the general idea of node selection that can be used in branch-and-bound, and they are not an add-on of SCIP. Therefore, performance improvements may still be expected by implementing and applying the existing algorithms to SCIP 8. If not, the rationale should be clearly provided.
> > >
> > > Because of the above concerns, I keep my score.

---

### Official Review · Reviewer_hSJj · 2023-11-02

**Soundness:** 2 fair
**Presentation:** 3 good
**Contribution:** 2 fair
**Rating:** 5
**Confidence:** 3

**Summary:**

The paper proposes a reinforcement learning framework to learning the node selection policy in branch-and-bound algorithm. In particular, the paper considers an environment based on SCIP. The considers a graph neural network on the branch-and-bound tree with root-to-leaf path aggregated scores as the policy net and employs policy gradient algorithms for training. The paper carefully generates TSP problems with moderate difficulty for training and evaluate the learned node selection policy on TSPLIB, UFLP, MINLPLib, MIPLIB. The results show that the learned node selection policy outperforms the default policy in SCIP in terms of Reward and Utility/Node.

**Strengths:**

* Most of the paper is well written and easy to understand for readers with basic knowledge in reinforcement learning and branch-and-bound.
* The root-to-leaf path aggregated score is a clever design. It avoids the computation challenge from the growing of the branch-and-bound tree by an intuitive assumption: if a node is good, so should be its ancestors.

**Weaknesses:**

* The definition of the reward is not rigorously defined. Specifically, the paper does not disclose how are the gap(node selector) and gap(scip) are calibrated. It could be
    * The gap when reaches the time budget.
    * The gap at the same number of nodes n, with $\text{traj}(\text{node selector})[:n]$ rolled out with node selector, $\text{traj}(\text{scip})[:n]$ rolled out with scip,
    * The gap at the same number of nodes n, with $\text{traj}(\text{node selector})[:n]$ and $\text{traj}(\text{scip})[:n-1]$ rolled out with node selector, $\text{traj}(\text{scip})[n-1:n]$ rolled out with scip.

* The score in evaluation need more justification. From my perspective, the most important goal of learning a node selection policy is finding good primal solutions. With this aim, none of the scores are a good choice. The second priority for a node selection policy is to close the duality gap. In this sense, "Utility/Nodes" is not good choices.

* The paper consider a small threshold 45 seconds. This is a relatively small time budget for solvers to solve MIP problems. To demonstrate the learned policy is practical, the results with a longer running time should be reported.

**Questions:**

* Only the results on benchmarks are provided in the paper. I am curious about the performance on the training data.

---

> ### Author Response · Authors · 2023-11-11
>
> First and foremost we want to thank the reviewer for their time and effort.
>
> > The definition of the reward is not rigorously defined. Specifically, the paper does not disclose how are the gap(node selector) and gap(scip) are calibrated. It could be
>
> Thank you, we did indeed not define this thoroughly enough.
> Your first variant is the correct one:
> We define the gap(node selector) and gap(scip) as the optimality gap reached by our selector or SCIP after the time budget is reached. This accounts for the fact that SCIP node-selection may be faster than our node selection, which wouldn’t be accounted for when measuring relative to the number of nodes solved. We updated the paper accordingly.
>
> > The score in evaluation need more justification. From my perspective, the most important goal of learning a node selection policy is finding good primal solutions. With this aim, none of the scores are a good choice. The second priority for a node selection policy is to close the duality gap. In this sense, "Utility/Nodes" is not good choices.
>
> The objective of our metrics was mostly to preserve the goal of minimizing the duality gap, while also accounting for the fact that the duality gap may have very different scales from problem to problem, hence the normalization using the SCIP baseline and zero-centering of the “our gap” / “scip gap” loss.
> We agree with you and the other reviewers, therefore we decide to also publish the shifted geometric mean of the duality gap for all instances in each dataset to give more of a global view of our methods performance. The reason we did not do so initially is due to the fact that shifted geomean does not work if infinite duality gaps are in the results. In the new results we model infinities as a duality gap of 10^20, just like SCIP would.
>
> > The paper consider a small threshold 45 seconds. This is a relatively small time budget for solvers to solve MIP problems. To demonstrate the learned policy is practical, the results with a longer running time should be reported.
>
>
> We also share your concern of our short runtime: We initially chose this runtime since we were mostly interested in the initial (most impactful) phase of node selection, and other works (such as Labassi et al.) only having an average solver time of less than thirty seconds, even on their largest instances.
> However, we do concede that evaluating with larger time budgets is very important: We have added additional runs on our real-world datasets with a time budget of 5 minutes for TSP, MINLPlib, and MIPLIB. For these tests we increase the number of nodes processed by our method from 250 to 650, as the longer overall runtime justifies running our selector for a longer time as well.
>
> As you can see, we still outperform the SCIP baseline despite only being trained on instances with a 45s time limit, usually even increasing the relative gap between our method and scip.
>
> > Only the results on benchmarks are provided in the paper. I am curious about the performance on the training data.
> We also added information on our training results in the appendix.
>
> We hope we managed to address all your concerns and are happy to discuss further.

---

### Author Response · Authors · 2023-11-15

Dear reviewers: we updated the paper with the previously described additional details and experiments.
Specifically from the point of view of new data, we added additional experiments for longer experiment runtimes (5 minutes) for all real-world benchmarks, and added the shifted-geometric mean of the optimality gaps as an additional comparison point.
To better highlight all changes, we marked them using red text.


We believe we have addressed all current concerns and are happy to engage in additional discussion!

---

### Meta-Review · Area_Chair_iqHQ · 2023-12-09

**Metareview:**

The paper introduces a reinforcement learning framework for node selection in the branch-and-bound algorithm for Mixed Integer Programming. While the paper exhibits some strengths in clarity, innovative design, empirical results, and code provision, several critical weaknesses lead to the recommendation for rejecting the paper.

A major concern is the lack of rigor in defining rewards, particularly in the absence of disclosed calibration methods for the gap (node selector) and gap (SCIP). Additionally, the choice of evaluation metrics, specifically "Utility/Nodes," lacks sufficient justification, raising questions about the reliability of the results.

The mathematical clarity of the reinforcement learning structure is criticized, with Reviewer KqYv highlighting a need for a more rigorous definition of the RL structure and calling for comparisons with state-of-the-art algorithms.

Reviewer rRw3 expresses concerns about the complexity and practicality of the proposed method, citing heuristic choices without clear justification. The need for addressing issues related to the choice of the upper bound factor, reward function design, and the use of PPO is emphasized.

Reviewer Lp54 raises significant concerns about the experimental setup, the lack of direct testing of key hypotheses, and clarity issues in justifying technical statements. The absence of comparisons with existing algorithms is viewed as a major drawback.

Overall, while the paper has some positive aspects, the highlighted weaknesses in rigor, clarity, experimental design, and comparison with existing methods contribute to the unanimous recommendation for rejecting the paper. Significant revisions are required to address these concerns and improve the overall quality of the submission.

**Justification For Why Not Higher Score:**

The current recommendation falls short of a higher score due to several critical concerns highlighted in the meta review. The lack of rigor in defining rewards, particularly the undisclosed calibration methods for the gap (node selector) and gap (SCIP), raises doubts about the reliability of the results. The choice of evaluation metrics, specifically "Utility/Nodes," lacks sufficient justification, further undermining the paper's overall credibility.

Additionally, the mathematical clarity of the reinforcement learning structure is criticized, with concerns raised by Reviewer KqYv regarding its definition. The absence of comparisons with state-of-the-art algorithms is viewed as a significant drawback, limiting the paper's contribution to the existing body of knowledge.

Reviewer rRw3 expresses concerns about the practicality of the proposed method, citing heuristic choices without clear justification. Issues related to the choice of the upper bound factor, reward function design, and the use of PPO are highlighted as areas requiring clarification and improvement.

Reviewer Lp54 raises substantial concerns about the experimental setup, the lack of direct testing of key hypotheses, and clarity issues in justifying technical statements. The absence of comparisons with existing algorithms is seen as a major limitation, contributing to the recommendation for rejecting the paper.

**Justification For Why Not Lower Score:**

N/A

---

### Decision · Program_Chairs · 2024-01-16

Reject